

# On the role of soil water retention characteristic on aerobic microbial respiration

Teamrat A. Ghezzehei[1], Benjamin Sulman[1], Chelsea L. Arnold[1], Nathaniel A. Bogie[1] Asmeret Asefaw Berhe[1]

[1]School of Natural Sciences, University of California, Merced, CA 95340, USA

*Correspondence to*: Teamrat A. Ghezzehei (taghezzehei@ucmerced.edu)

**Abstract.** Soil water status is one of the most important environmental factors that control microbial activity and rate of soil organic matter decomposition (SOM). Its effect can be partitioned into effect of water energy status (water potential) on cellular activity, effect of water volume on cellular motility and

aqueous diffusion of substrate and nutrients, as well as effect of air content and gas-diffusion pathways on concentration of dissolved oxygen. However, moisture functions widely used in SOM decomposition models are often based on empirical functions rather than robust physical foundations that account for these disparate impacts of soil water. The contributions of soil water content and water potential vary from soil to soil according to the soil water characteristic (SWC), which in turn is strongly dependent on soil texture

and structure. The overall goal of this study is to introduce a physically based modelling framework of aerobic microbial respiration that incorporates the role of SWC under arbitrary soil moisture status. The model was tested by compariing it with published datasets of SOM decomposition under laboratory conditions.

## 1 Introduction

Soil moisture is one of the primary physical factors that control microbial activity (Harris, 1981). Short- and long-term temporal variations in soil moisture are strongly correlated heterotrophic respiration rates (Carbone et al., 2011; YUSTE et al., 2007). Therefore, the moisture-decomposition relationship is an important determinant of geographic distribution and climatic sensitivity of soil organic carbon (SOC) stocks (Moyano et al., 2013; Schmidt et al., 2011). The microhabitats that influence the community

structure and activity of soil microbes (Tecon and Or, 2017) are far too small compared to the macroscopic

measures of average soil water status; such as volumetric water content, relative saturation or water holding

capacity. At pore and sub-pore scales, the volume and connectivity of water pools and films is dependent

on matric potential—a measure of the strength by which water is held in pores and on surfaces. Matric

potential determines the thickness of water films (on very dry soils), curvature of the capillary menisci, and

the largest drained pore-throat. The relationship between the bulk soil water content and the average matric

potential—commonly referred to as soil water characteristic (SWC) or water retention curve (WRC)—is a

macroscopic measure of hydrologically relevant pore-size distribution and surface area (Hillel, 1998). As

such, it is also a reflection of soil texture, which controls surface area and pore size distribution, and

structure, which controls total porosity, and abundance of intra- and inter- aggregate porosity.

In process-oriented mathematical models of soil organic matter (SOM) dynamics (Coleman and Jenkinson,

1996; Parton et al., 1998), sensitivity of SOM decomposition to soil moisture is often modelled in terms of

functions that scale the maximum decomposition rate as a function of volumetric water content (Sulman et

al., 2012). Optimal decomposition rate has been shown to peak at or near *field capacity* (defined

interchangeably as matric potential of -30 kPa or water content after a saturated soil is drained for 24-48

hours) with significant reductions in decomposition towards the wet and dry ends of soil moisture range

(Franzluebbers, 1999; Linn and Doran, 1984; Monard et al., 2012; Sierra et al., 2017; Tecon and Or, 2017).

Typically, such bell-shaped soil moisture sensitivity curves are described using dimensionless polynomial

scalars that are calibrated against experimental data (Sulman et al., 2012; Wickland and Neff, 2007).

Skopp et al., (Skopp et al., 1990) proposed one of the earliest conceptual models that attempted to provide

mechanistic rationale for why decomposition of SOM exhibits peak rate at certain water content in terms

of balance between substrate diffusion and gas diffusion. The model describes aerobic respiratory activity

as a process limited by gaseous diffusion and/or aqueous diffusion, at the wet and dry ranges of soil moisture

spectrum, respectively,



$$P = \min \begin{cases} \gamma D_N(\theta) \\ (1-\gamma)D_O(\theta) \end{cases} \tag{1}$$

where $P$ is an index of decay rate, $\gamma$ is the relative weight (importance) of aqueous diffusion of nutrients, and $D_N$ and $D_O$ are water content ($\theta$) dependent effective diffusion coefficients of nutrients and oxygen, respectively. This model, which results in an inverted 'V'-shaped curve, has sufficient flexibility to capture

results from lab incubation experiments. Beyond bulk OM dynamics, this model formulation was shown to capture how nitrification rate of texturally contrasting soils correlate with gas diffusivity under high water content (Schjønning et al., 2003; 2011). Furthermore, the model has been able to capture observed increases in decomposition rate with water content (hence, aqueous diffusion) (Franzluebbers, 1999; Linn and Doran, 1984; Miller et al., 2005; Thomsen et al., 1999).

However, the direct influence of water potential on microbial activity and decomposition rate has not been widely adopted in SOM dynamics models (Moyano et al., 2013; 2012). In aqueous media, microorganisms respond to osmotic stress (low osmotic potential) by accumulating electrolytes and small organic solutes that counter the water potential gradient across their membranes (Wood, 2011). The resulting high intracellular osmotic potential inhibits production and activity of enzymes in bacteria (Csonka, 1989;

Skujins and McLaren, 1967) as well as fungi (Grajek and Gervais, 1987; Kredics et al., 2000). In unsaturated soils, microorganisms are additionally subjected to matric potential of water, which is comprised of adsorption of thin films on mineral surfaces and capillary attraction of menisci (Hillel, 1998). Thus, enzymatic activity, community composition, and overall activity of bacteria and fungi inhabiting unsaturated soils are significantly impacted by both concentration of dissolved solutes (osmotic potential)

and reduced water content (matric potential) (Chowdhury et al., 2011a; 2011b; Manzoni and Katul, 2014; Stark and Firestone, 1995; Tecon and Or, 2017). It is important to note that soil drying concentrates solutes in pore water, further reducing osmotic potential. However, because water content and matric potential are strongly correlated through the SWC, their effects on microbial respiration and decomposition of SOM are often lumped together or considered interchangeable (Moyano et al., 2012; Sierra et al., 2017).

Unless empirical moisture sensitivity curves are calibrated individually for each soil, ignoring the independent contributions of water potential and water content on microbial activity is tantamount to discounting the role of soil texture and structure on soil-moisture sensitivity curves. This drawback is especially critical in land surface models that might be applied across many different soil types. In long-

term simulations of land-surface processes, the feedback of changes in SOM stocks on soil aggregation and structure—hence, SOM decomposition rate—may not be accurately captured if the effects of water content and water potential are lumped together. It is also an important limitation in modelling SOM dynamics in soils that undergo drastic structural change over short period of time; e.g., via tillage or slaking of dry aggregates during rapid rewetting.

The objective of this study was to provide a modelling framework that allows integration of SWC in SOM dynamics modelling. We introduce conceptual and mathematical model of SOM dynamics that accounts for the independent role of soil aeration, water content, and water potential. For simplicity, we limit our analysis and illustration of the model to a single pool of SOM under isothermal conditions. However, the framework can be readily expanded to multiple-pools and dynamic thermal regime.

**2 Materials and Methods**

Process based SOM dynamics models provide conceptual basis for quantitatively describing the biophysical interactions within soil system that determine the fate of SOM. However, the model parameters that represent soil and SOM properties and biophysical rates are difficult to determine *a priori*. Thus, these parameters must be extracted from experimental data via inverse modelling (fitting). Whether the fitted

parameters retain their physical significance when the models are applied to contexts and scales that are not represented in the experimental data is a major challenge for most predictive modelling applications (Finsterle and Persoff, 1997). The pitfalls in this regard include strong correlation between fitted parameters and over-fitting of experimental data (fitting of random errors at the expense of retaining the ability to

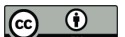


generalize). These pitfalls can be partially avoided by reducing the number of tuneable free parameters and/or determining some of the parameters independent of the experimental data that is to be fitted.

The overall goal of the model proposed in this study is to incorporate the role of SWC in modelling of SOM dynamics under arbitrary soil moisture status. To achieve this goal in a robust and generalizable manner,

we chose to represent SOM dynamics using a simple single-pool, first-order kinetics. This model relies on only two parameters: the size of the active SOM pool and a constant decay rate. The effect of soil water status and SWC are incorporated in these parameters by relying on well-established relations of multiphase flow and transport concepts and independently fitted SWC curves. This was done without adding new free parameters that are tuned to fit observed SOM decomposition data.

**2.1 Soil Water Characteristic**

Soil-water characteristic is a constitutive relationship between the soil volumetric water content and matric potential. It embodies the pore-size distribution and as such is a quantitative representation of soil texture and structure. It exerts direct control on macroscopic and microscopic water content distribution, and indirectly influences flow of water, transport of dissolved constituents and gas fluxes. It also has strong

bearing on the activity of soil microorganisms and plant roots. SWC is also sensitive to changes in soil structure. The wet end of SWC readily responds to changes in bulk density (e.g, tillage and compaction, root and macro fauna activity, freezing and thawing, drying and rewetting) (Aravena et al., 2013; Ghezzehei, 2000; Or et al., 2000; Ruiz et al., 2015).

SWC is typically represented by monotonic sigmoid function, the most common being van Genuchten's

(van Genuchten, 1980) equation

$$\Theta = (1 + (\alpha\psi)^n)^{-m} \tag{2}$$

where $\Theta = (\theta - \theta_r)/(\theta_S - \theta_r)$ is effective water saturation; $\theta$, $\theta_r$, and $\theta_S$ are volumetric water content, residual water content, and saturated water content, respectively; $\psi$ [kPa] is matric potential; $\alpha$ [kPa] is a





parameter that indicates the matric potential at which maximum drainage of soil water occurs; and $n$ $(1 < n < \infty)$ and $m = 1 - 1/n$ are shape parameters that reflect the spread of the SWC function. Matric potential can be related to an effective pore-throat diameter using Young-Laplace law as $D \approx 4\sigma/\psi$, where $\sigma$ [N m$^{-1}$] is surface tension of pore water. Therefore, the SWC function (2) can re-written in terms of the

pore-throat diameter as,

$$F = \left(1 + \left(\frac{D_0}{D}\right)^n\right)^{-m} \tag{3}$$

where $F = \theta/\theta_S$ represents the relative saturation or cumulative pore size distribution. Eq. (3) is re-interpretation of SWC as cumulative pore size distribution and $D_0 \approx 4\alpha\sigma$ stands for the modal pore-throat diameter. The pore-throat diameter scale is shown on the top axis of the figure. In Fig 1, Eq (2) and (3) are

illustrated by the solid line. The corresponding pore-size density function $f = dF/dD$ is shown as the shaded bell-shaped curve. This form of SWC is a good approximation for soils with unimodal pore-size distribution.

However, soils with significant level of aggregation, clumping and/or biopores exhibit multimodal pore size distributions—for example with fine intra-aggregate pores and coarse inter-aggregate pores. Such soils

can be represented by summation of two or more uniomodal pore-size distributions. For soils that exhibit bimodal pore size distribution. by sums of two van Genuchten curves (Durner, 1994)

$$\Theta = \sum_{i=1}^{2} w_i (1 + (\alpha_i \psi)^{n_i})^{-m_i} \tag{4}$$

where $w_1 + w_2 = 1$ represents the relative weights of the inter- and intra- aggregate pore populations.

It is important to note that water retention is dominated by capillary attraction in the wet end of the SWC

curve, approximately $\psi > -10^{-2}$ kPa and $D > 1$ µm, while adsorption of thin water film on mineral surfaces dominates in the dry range (Or and Tuller, 1999). Thus, soil texture is the most important



determinant in the dry end of SWC while structure and water-stable aggregation dominate in the wet end. The latter is strongly influenced by amount and nature of SOM, and readily responds to changes in SOM content.

**2.2 SOM Dynamics Modelling**

The conceptual basis for our model is that soil organic matter is comprised of a single pool characterized by first-order rate of decomposition

$$\frac{dC}{dt} = -\kappa C \tag{5}$$

where $C$ [mg-C/g-SOC$_0$] is the active C pool remaining at any given time, expressed as a fraction of the total initial SOC and the rate constant $\kappa$ [day$^{-1}$] is a measure of SOM decomposition largely driven by living

decomposers. Therefore, we consider it to be a composite parameter that accounts for the abundance of decomposer population as well as the activity of an average decomposer. Both of these factors are impacted when soil moisture level changes. (Chowdhury et al., 2011b) observed that the abundance of active decomposers declines while maintaining the same level of average activity as water potential dropped from $\psi = 0$ kPa to $\psi = -2000$ kPa. Organisms subjected to low total water potential exhibit reduced population

growth as substantial proportion of their energy intake is routed towards osmo-regulation (Harris, 1981; WATSON, 1970). Upon further drying, however, the population remained constant but the activity declined sharply (Chowdhury et al., 2011a; 2011b). Previously, (Stark and Firestone, 1995) used two independent techniques to evaluate the relative importance of water potential on cytoplasmic dehydration and the role of water content diffusional limitations in controlling rates of nitrification in soil. Nitrification rates in well

mixed soil slurries, in which NH$_4$ was maintained at high concentrations and osmotic potential was controlled by the addition of K$_2$SO$_4$, declined exponentially with reduction in water potential (0 to ~ -4000 kPa) of the slurries. In a companion moist soil incubation experiment, in which substrate supply was controlled by the addition of NH$_3$ gas, they observed that steeper decline in nitrification as a result of combined effects of reduced diffusion and cytoplasmic dehydration. Similarly, (Tresner and Hayes, 1971)




showed that in the absence of diffusion limitation the survival probability of fungi declines with water

potential. In the proposed model we assume that the diffusion limitation does not directly control the rate

constant. But rather, its effect on SOM decomposition rate ($dC/dt$) through its impact on the accessibility

of (Davidson et al., 2012).

Another moisture related factor that impacts decomposition rate constant by aerobic processes is availability

of dissolved $O_2$ in pore water. Because diffusion of aqueous $O_2$ is seven orders of magnitude slower than

that of gaseous $O_2$, gas diffusivity is the primary factor that indicates $O_2$ limitation in SOM dynamics (Skopp

et al., 1990). (Schjønning et al., 2003) compared nitrification rate of cores sampled from three soils of

contrasting textures and equilibrated at seven matric potential levels, -0.015 to 1.5 kPa, near the wet end of

the moisture spectrum. They observed nitrification rates increased in all soils as water content was reduced

from saturation, and then decreased with further decline in water content. The initial increase was not

correlated with water content or matric potential. However, consistent with the model of (Skopp et al.,

1990), relative gas diffusivity was a good predictor of nitrification.

Based on the above observations, we propose to expand the decomposition rate $\kappa$ in to product of multiple

interacting components that represent biophysical factors,

$$\kappa = \kappa_o \prod_i \kappa_i \qquad (6)$$

where $\kappa_i$ [0,1] are dimensionless constants representing the biophysical factors. Here we focus only on the

limiting effects of water potential and available dissolved oxygen. The parameter $\kappa_o$ [day$^{-1}$] is an intrinsic

(maximum) rate constant and represents lumped effect of all the remaining unresolved biophysical factors

such as temperature, pH, soil mineralogy, OM composition, and nutrient availability. In principle, Eq. (5)

can be expanded to accommodate as many variables as needed. This general formulation has been used to

represent the effects of various enzyme activities and temperature (Sierra et al., 2017).




Here we propose an exponential equation to describe the dependence of soil microbial activity on water potential,

$$\kappa_\psi = e^{\lambda\psi} \tag{7}$$

where $\lambda$ [kPa$^{-1}$] is a factor that represents the dependence of respiration rate on matric potential. This trend

is assumed to account for the decline in population of decomposers as well as reduced per capita activity at very low water potentials. The model fits well the trend of nitrification in slurries observed by (Stark and Firestone, 1995) ($\lambda = 5.8 \times 10^{-4}$ kPa$^{-1}$) and the survival probability of fungi in the absence of diffusion limitation observed by (Tresner and Hayes, 1971) ($\lambda = 7.58 \times 10^{-5}$ kPa$^{-1}$). Here we use the geometric mean of these two coefficients ($\lambda = 2.1 \times 10^{-4}$ kPa$^{-1}$) to account for the fact that both bacteria and fungi

are involved in soil respiration and that nitrification is more sensitive to resource limitation than respiration (Schjønning et al., 2003; Scott et al., 1996). Comparison between the proposed trend and dimensionless nitrification data of (Stark and Firestone, 1995) is shown in Fig 2c. The steepest decline in effective microbial activity occurs in the range $-10^4 \leq \psi \leq -10^2$ kPa. Note that although the primary state variable in Eq. (6) is matric potential, it is tacitly assumed that the equation also accounts for decrease in osmotic

potential that accompanies concentration of solutes in drying soils (Chowdhury et al., 2011b).

Following (Skopp et al., 1990), we assume the relative dependence of SOM decomposition on dissolved $O_2$ can be explained by the relative gas-phase diffusivity, which in turn is inversely correlated with tortuosity of the gas phase,

$$\kappa_a = \frac{D_g}{D_{g,0}} \propto \frac{1}{\tau} \tag{8}$$

where $D_{g,0}$ and $D_g$ are diffusivities in open air and soil, respectively, and $\tau$ is tortuosity. Here we use the well-known, parameter free Bruggeman expression for tortuosity $\tau = a^{1/2}$, where $a = \phi - \theta$ is air-filled porosity (Pisani, 2011). However, this model does not account for the distance from air-exposed soil surface. In lab incubation studies, short cores and/or cores with large exposed surfaces do not exhibit



significant $O_2$ limitation as the average diffusion distance is short. Conversely, in field conditions, $O_2$ availability becomes increasingly limiting with depth as transport length increases and cumulative $O_2$ consumption increases (Angert et al., 2015). Therefore, we add a correction term that accounts for these variations

$$\kappa_a = \kappa_{a.\min} + (1 - \kappa_{a.\min})\left(\frac{\phi - \theta}{\phi}\right)^{1/2} \tag{9}$$

The parameter $\kappa_{a.\min}$ represents the minimum relative SOM decomposition rate when the soil is fully saturated and the $O_2$ limitation is at its peak. A value of unity implies no $O_2$ limitation whatsoever and corresponds to very shallow soil. On the other hand, small values of $\kappa_{a.\min}$ are applicable for deeper soils and/or longer cores. Further controlled experiments are needed to ascertain how this parameter varies with

10 depth or sample configuration. The effect of $\kappa_{a.\min}$ on the overall trend of the relative decomposition rate is shown in Fig 2a.

Another mechanism that water content exerts control over SOM decomposition is through its effect on substrate accessibility to decomposer microorganisms., Aqueous phase diffusivity of soluble substrates becomes increasingly limited as liquid phase connectivity is reduced and transport distance increases

(Moldrup et al., 2004; Skopp et al., 1990). We assume the fraction of active SOC pool that is accessible to decomposers scales with relative aqueous diffusivity, which is modelled using Bruggeman expression for tortuosity of the liquid phase

$$\frac{C_A}{C} = D_w = \left(\frac{\theta}{\phi}\right)^{1/2} \tag{10}$$

where $C_A$ stands for the fraction of the active pool of SOC that is accessible to decomposers at the ambient

moisture level (Fig 2b). When the soil pores are saturated with water, the active pool is accessible in its entirety. Additionally, it is possible to experience reduction of the absolute quantity of substrate in aqueous phase solution as the increased concentration of dissolved substrates induces sorption (complexation with



mineral surfaces) (Šimůnek et al., 2016). This latter effect, which requires inclusion of reactivity of the

mineral surfaces, is not incorporated in this study but can be readily added if the requisite properties of the

solid phase and SOM are known.

The SOM dynamics under arbitrary fluctuation of soil water status (i.e., $\theta(t)$ and $\psi(t)$) can be described

by solving Eq. (5) subject to initial active pool of SOC, $C(t = 0) = C_0$,

$$C(t) = C_0 \exp\left( \kappa_o \int_0^t K(\theta,\psi)d\tau \right) \tag{11}$$

where $K(\theta,\psi)$ is moisture sensitivity function derived by combining modifiers that represent effects of

matric potential (Eq. 7), $O_2$ diffusion (Eq. 9) and accessibility of SOM (Eq. 10),

$$K(\theta,\psi) = C_0 \exp\left( \kappa_{a.min} + (1 - \kappa_{a.min})\left(\frac{\phi - \theta}{\phi}\right)^{1/2} \right) \tag{12}$$

Moisture sensitivity calculated using a typical unimodal SWC is illustrated in Fig 2d. At steady water

content and water potential, the integral can be evaluated analytically leading to a simple closed-form

solution,

$$C(t) = C_0 \, e^{-\kappa_o K(\theta,\psi) t} \tag{13}$$

These solutions have only two free parameters, which are not dependent on water content: initial fraction

of the active pool $C_0$ and the maximum decay rate $\kappa_o$. Water content and matric potential are linked via the

appropriate SWC equation (Eq. 2 or Eq. 3). Variations in SOM decomposition between different water

content levels are explained by independently determined SWC. It is important note here that

characterization of SWC has become more accessible in the past decade with the introduction of apparatus

that rely on evaporation rather than regulated pressure (Schindler et al., 2010). Moreover, pedotransfer



functions that predict SWC parameters from routinely measured soil properties (e.g., texture, bulk density and SOM) are becoming increasingly more reliable (Zhang and Schaap, 2017)

For comparison with incubation experiments, cumulative $CO_2$-C evolution can be evaluated by subtracting the dynamic SOC content  (Eq. 10 or Eq. 11) from the initial active stock.

$$C_{CO_2}(t) = C_0 - C(t) \qquad (14)$$

where $C_{CO_2}$ stands for the cumulative evolved C expressed as fraction of the initial SOC.

### 2.3 Data for Model Testing

Testing the validity of the model in simulating SOM dynamics requires cumulative $CO_2$-C evolution data from incubation experiments conducted at multiple constant water content levels as well as knowledge of concurrent water content and matric potential values. We obtained laboratory incubation data that meet these requirements, comprising 31 soils, from four published sources. These soils span a wide range of textural classes, SOM concentrations, and soil structural states. Three of the studies were from experiments conducted at steady wetness level and one is from a study involving drying and episodic rewetting. Summary of the datasets used is given in Table 1. The datasets used are described briefly below. The fact that none of the datasets include fully saturated soil is recognized as drawback in the present state model validation.

**Arnold et al** (2015): incubated soils from high elevation meadows in the Sierra Nevada, California, at five different water potentials (-10 to -400 kPa) and measured the $CO_2$ efflux 11 times over 395 days. Soil samples were collected from three distinct hydrologic regions within the meadow area (wet, intermediate and dry) at three depths. SWC data were collected on separate samples using pressure-plate apparatus, which were fitted with bimodal SWC model of (Durner, 1994). The best-fit SWC curves were used to estimate the water content levels of each treatment.



**Franzluebbers** (Franzluebbers, 1999): collected samples from the surface (0-10 cm) of 15 variably eroded soils of the Madison-Cecil-Pacolet, near Farmington GA. Samples were packed into bottles at two bulk density levels: naturally-settled and lightly-compressed. The resulting 30 distinct soils were incubated at eight water content levels and $CO_2$ efflux was measured three times over incubation period of 24 days.

Matric potential of the samples were measured at the end of the incubation experiment by the filter-paper method. A digitized version of this dataset was published as supplemental material by (Moyano et al., 2012).

**Don** (Moyano et al., 2012): additionally, a previously unpublished dataset set by A. Don, that included a 30-day incubation of one soil at five water content levels was obtained from supplemental dataset published

by (Moyano et al., 2012). $CO_2$ efflux data was provided hourly. Matric potential values were inferred from a unimodal SWC curve (van Genuchten, 1980) that was estimated using the pedotransfer function ROSETTA (Schaap et al., 2001).

**Miller et al** (Miller et al., 2005): performed a laboratory incubation to evaluate the impact of short-term fluctuations in soil moisture on long-term carbon and nitrogen dynamics. The study was designed to mimic

seasonal wetting of dry soils that is characteristic to many arid and semi-arid environments. Sandy clay loam soil samples collected from Sequoia National Park, with C concentration of 2.3%, were incubated in centrifuge tubes. The tubes were wetted to 60% water holding capacity (WHC) and then allowed to dry by evaporation until they were due for rewetting treatment. WHC was defined as the gravimetric water content of saturated soil allowed to drain for 6 hours. Four and two week of rewetting intervals were tested over a

16 week incubation period. Daily $CO_2$ efflux and water content (expressed in terms of WHC) were provided. The corresponding matric potential values were inferred from a unimodal SWC curve (van Genuchten, 1980) representative for the textural class (Schaap et al., 2001).





### 2.4 Fitting of Model to Data

The first step of fitting the model to experimental data involves calculating the concurrent water content

and matric potential levels at all times as described above. For each of the unique soil types considered, the

cumulative $CO_2$ efflux data from all the different water content levels were fitted together by optimizing

initial fraction of the active pool $C_0$ and the maximum decay rate $\kappa_\circ$, using non-linear Levenberg–

Marquardt algorithm implemented in the **minpack** package (Elzhov et al., 2016) of R (R Core Team, 2017).

For all the soils used in this study, the parameter that represents $O_2$ limitation in saturated soils was set to

$\kappa_{a.\,\min} = 0.2$. Validity of this estimate and its sensitivity to soil depth and soil type needs further

investigation.

**3 Results**

Simultaneously measured water content and matric potential data from the studies of Arnold et al. and

Franzluebbers (Arnold et al., 2015; Franzluebbers, 1999) along with the best-fit bimodal and unimodal

SWC curves are reported in Figs. 3 and 4, respectively. The best SWC parameters of all the soils used in

this study are reported in Table A1. The SOM-rich meadow soils of Arnold et al. (2015) were developed in

cold, high-altitude environment where estimated annual input of SOM far exceeds decomposition. In these

soils, SOM content and porosity decrease with depth in all three hydrologic regimes. SOM and porosity

across the three sites are ranked as wet >intermediate>dry. All the meadow soils studied exhibit two distinct

pore size classes representing (a) large pores between decomposing fibers of organic matter (in the surface

peats) and between aggregates (in the subsoils) and (b) finer pores between processed SOM and mineral

fractions. The macropores of these soils drain when subjected to low suction (approx. -5 kPa). However,

the soils remain fairly wet until they are subjected to matric potentials lower than approx. -300 kPa.

The mineral soils in contrast, exhibited unimodal SWC (Franzluebbers, 1999). The compressed samples

had slightly lower porosity than their naturally settled counterparts, across all textures investigated. The

water content decreased continuously as the matric potential was lowered progressively. However, the

compressed soils needed lower matric potential to drain to the same level of wetness. This indicates that compression caused the pores to shrink across most of the pore-size distribution.

The model proposed in this study suggests that sensitivity of SOM decomposition to soil moisture dynamics is explained in its entirety by the SWC, which represents concurrent states of air content, water content and

matric potential. Moisture sensitivity curves of all soils calculated using as Eq. (11) are depicted in Fig. 5. The difference between the soils with unimodal and bimodal SWC curves is mostly reflected in the water potential range for peak decomposition. In addition, compaction results in shift of the moisture sensitivity curves to the dry end, which is a reflection of reduced of mean pore size.

Temporal $CO_2$ evolution data for a subset of meadow soils (0-10 cm) are compared with best-fit model

simulations in Fig 6. We assumed compaction does not alter the optimal decay rate and active pool. Thus, the datasets from the naturally settled and compacted samples were fitted with common parameters. As indicated above, only the initial fraction of the active pool $C_0$ and the optimal decomposition rate $\kappa_\circ$ were optimized for each of the soils. The complete set of best-fit plots and fitted parameters are given Fig A2. For the mineral soils of Franzluebbers (Franzluebbers, 1999), the final SOC loss during 24-day incubation

are compared with model fits in Fig 7. The corresponding temporal $CO_2$ evolution data and best-fit model simulations for all the mineral soils are depicted in Fig A3. Bulk density levels of individual samples of the same soil that were incubated at different levels of matric potential were not consistent. Bulk density of individual samples are indicated within each plot subpanel in Fig A3. As a result, the differences between compacted and naturally settled samples were not consistent across the matric potential spectrum. However,

the SWC curves were fitted to the soil water content and matric potential, by ignoring these inter-sample heterogeneities. The mismatch between measured and simulated $CO_2$ evolution include this discrepancy. Temporal $CO_2$ evolution data and best-fit model simulations for all the mineral soil of Don (Moyano et al., 2012) are depicted in Fig S4. The best-fit model parameters for all the soils are provided in Table A1.

The best-fit optimal decay rates for all the steady moisture experiments are plotted against SOC, active

SOC pool $C_0$, and incubation period in Fig 8. Recall that the duration of the incubation experiments of

Franzluebbers (Franzluebbers, 1999) and Don (Moyano et al., 2012) were much shorter than that of Arnold

et al. (2015) (24 and 31 days vs 395 days, respectively). Comparing Fig 8b and 8c suggests that the fraction

of the SOC stock involved in decomposition (size of the active pool) increases with incubation period. This

is to be expected as longer incubation period allows pools with slower decay rate to contribute at an

observable rate. Therefore, the average decay rate decreases with incubation period (Fig 8c), as the model

used in this study considers only one pool. The apparent correlation between the fitted parameters (Fig 8b)

is partially explained by this phenomenon as well.

Finally, comparison of the measured $CO_2$ evolution data from all the three studies (1375 data points

representing 40 different soils) are compared with the model fits in logarithmic scale and linear-scale (inset)

in Fig 9. The colour intensity of the points reflects density of data points. Over all, the model is in excellent

agreement with experimental observations across the full range of measured data.

Comparison between $CO_2$ evolution data of Miller et al. (Miller et al., 2005) under drying and rapid-wetting

condition with model simulations are shown in Fig 10. The fluctuation in the $CO_2$ evolution rate is explained

by the dynamics of water content (Fig 10a) and matric potential. Because a closed-form solution does not

exist for arbitrary fluctuations of soil moisture, the integral in Eq. (10) was evaluated numerically. Two sets

of model fits were performed. In the first, data from the two- and four-week rewetting intervals were fitted

together using one set of initial fraction of the active pool $C_0$ and the optimal decomposition rate $\kappa_\circ$ (Fig

10b). However, as shown in Fig 10, the two intervals started with a distinct difference at the initial

measurement period, which is assumed to reflect significant inter-sample difference. Therefore, a second

model fit was conducted, by treating the two intervals separately (Fig 10c).

## 4. Discussion and Conclusions

In the remainder of the discussions, soil matric potential is considered as the primary independent state variable, while water content and decomposition modifiers are all functions that depend on water potential. For all the soils investigated, the peak decomposition rate was approximately 60% (Fig 5) of the optimal

rate that would occur if aqueous diffusion, gaseous diffusion and water potential were not limiting. However, in soils where one or more of these factors are limiting across the spectrum of possible moisture range, SOM decomposition occurs under a suboptimal rate. The individual contributions of these limiting factors are shown in Fig A1. The effect of water potential is assumed to be due to matric potential only. This assumption ignores increase in solute concentration during drying and associated decrease in matric

potential. The limiting effects of aqueous and gaseous diffusion directly depend on water and content and porosity, therefore depend on SWC.

Soils with a broad range of pore size distribution drain incrementally over a wide range of matric potential, thus maintaining broad range of favourable moisture status. This is clearly demonstrated in the contrast between the moisture sensitivity of the meadow soils and the rest of the soils. Most of the meadow soils

show peak decomposition in the between −1000 kPa and −10 kPa, with rapid drop in decomposition under saturated condition. Recall that the minimum effective rate for saturated soils varies with $\kappa_{a.\,min}$, which reflects distance from the soil surface (see Fig 2a). The value of this parameter is likely to be lower in field conditions than for experimental cores. The rest of the mineral soils exhibit peak decomposition over narrow range of matric potential. The peak for the latter generally occurs at moisture level wetter than field

capacity. Compression of the mineral soils studied by (Franzluebbers, 1999) lowered the matric potential at which peak rate occurs. This is to be expected as compression reduces the pore sizes thereby decreasing the matric potential needed to drain the pores.

### 4.1 Implications

Knowledge of controls on soil C dynamics has improved in recent years and the focus has switched from

predominantly molecular level controls on SOM decomposition/stability, to a broader recognition that





environmental and physical conditions are more important controls on persistence of SOM. While the influence of temperature on SOM decomposition has received considerable attention, water remains the primary variable that confounds our ability to predict how soils in all climate zones will respond to perturbations both human-induced or naturally caused (Wieder et al., 2017). This model provides a first

step to bridging that gap (Kleber, 2010; Schmidt et al., 2011). The model has been applied to a wide range of soil types highlighting the critical but underrepresented role that soil structure and water play. Results shown in Fig 5 suggest that peat soils, once drained below a threshold, are prone to rapid loss of SOC over wide range of water potential, as their bimodal pore size distribution allows them to retain sufficient moisture to promote microbial activity. The effect of warming on increasing microbial activity and rapid C

loss from cold high-altitude and high-latitude environments has received considerable attention in recent years (Wieder et al., 2017). SOM in these regions has been protected in part by anoxic conditions. The model proposed here suggests these soils are prone to accelerated loss of SOM due to the extended water potential range for peak decomposition afforded to them by virtue of their pore-structure. This hypothesis has yet to be tested (Ise et al., 2008).

The above observations also show the importance of dynamics of the physical structure of soils (e.g., tillage or slaking) in regulating SOM dynamics. For example, this model suggests that disturbance of aggregated soils initially promotes rapid mineralization by widening the pore size distribution. This mechanism is in addition to the oft-credited liberation of SOM protected inside soil aggregates. However, with repeated wetting-drying cycles the soil structure is restored to its pre-tillage state by slaking of aggregates or

reconsolidation by capillary forces (Ghezzehei and Or, 2000; Liu et al., 2014; Or et al., 2000). Therefore rapid loss of C in tilled soils is likely to be short-lived. If true, this self-limiting phenomenon is likely to have had a beneficial effect in pre-industrial agriculture, when crop nutrition was derived by recycling of SOM. High demand for nutrients during the early season is matched by rapid mineralization, while a slow down later in the season protects SOM for subsequent seasons. To address these effects of soil structure

dynamics, it is important to incorporate the effect of soil structure in SWC.



The assumptions underlying the proposed model need to be tested and evaluated for wide range of soil environments. It is likely that sensitivity to water potential varies across soil types and the specific microbial communities. Therefore, variations of the slope of the water potential sensitivity curve $\lambda$ across soil types and environments needs to be evaluated. Contribution of salinity to total water potential is not accounted

5    for here. Provided that total solute concentration remains constant, it is possible estimate the dissolved fraction and its osmotic potential using sorption-desorption isotherms. However in soils that regularly receive considerable salt inputs (e.g., saline irrigation water, fertilizers, atmospheric depositions), complete solute balance consideration is necessary.

In summary, the proposed model opens a new way of interrogating the effect of soil structure, structural

10   dynamics and hydrologic processes on SOM dynamics. It is particularly valuable tool that can support formulation of testable and quantitative hypotheses. With proper calibration and testing, this model has the potential of filling much needed coupling between biogeochemical cycling and soil hydrology over wide range of temporal and spatial scales.





**Tables**

Table 1.

| Study | Arnold | Don | Franzluebbers | Miller |
|---|---|---|---|---|
| **Number of soil types** | 9 | 1 | 15 x 2 | 1 |
| **Water content levels** | 5 | 5 | 8 | 4 |
| **CO2 efflux measurements** | 11 | 100? | 3 | 1 |
| **Incubation duration (days)** | 395 | 1 | 24 | 110 |
| **Incubation temperature °C** | 20 | 21 | 25 | lab |
| **SWC type** | Bimodal | Unimodal | Unimodal | Bimodal |





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





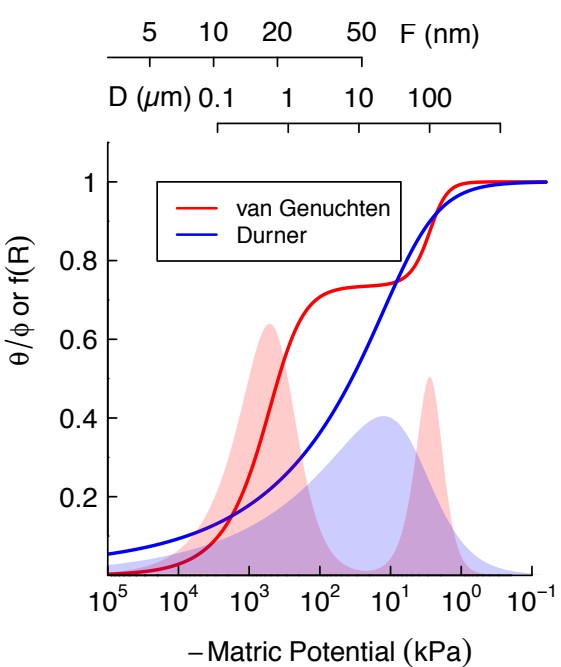

Figure 1: Schematic comparison of unimodal vs biomodal soil water characteristic (SWC) curves, represented using van Genuchten (1980) and Durner (1994) models, respectively. Shaded regions are distribution functions of effective pore throat diameter. Scales on top show the thickness of adsorbed film and pore-throat dimeter corresponding to the water potentials.





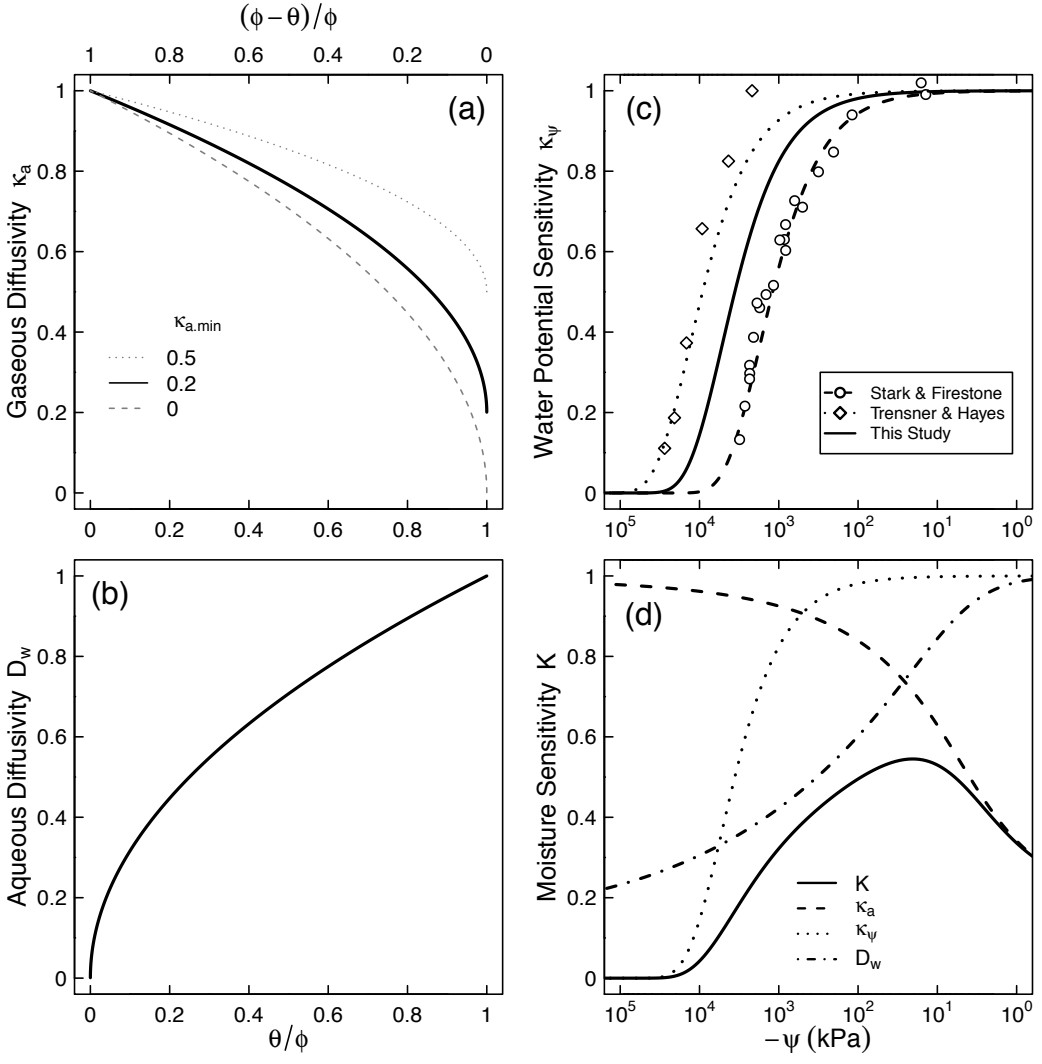

Figure 2: Relative contributions of (a) air diffusion on access to $O_2$, (b) limiting effect of water potential on microbial activity, and (c) aqueous diffusion limitation on substrate access. The combined effect of the three factors for a soil characterized by a unimodal SWC curve shown in Figure 1.




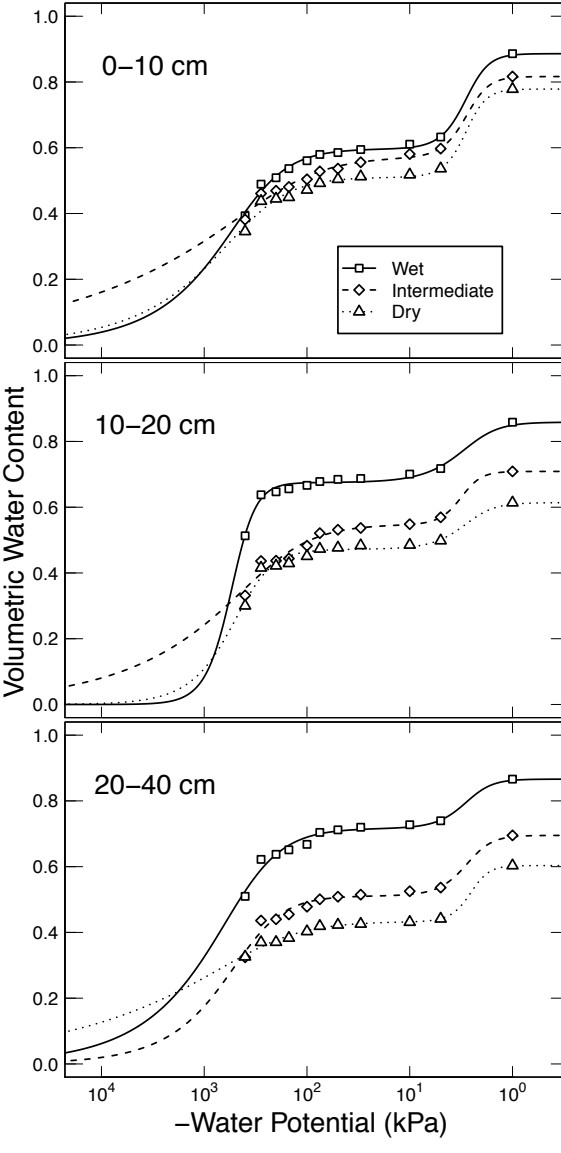

Figure 3: Water retention characteristics of meadow soils (Arnold et al, 2014) that were used to derive the relative effect of water potential on overall mineralization rate.





Figure 4: Soil moisture characteristics of soils analyzed by Franzluebers (1999); symbols are measured values and lines are van Genuchten model fits. The best fit $n$ parameter are shown. Soils at natural (triangle symbol and dashed line) and compacted (circle and solid lines) state were studied.





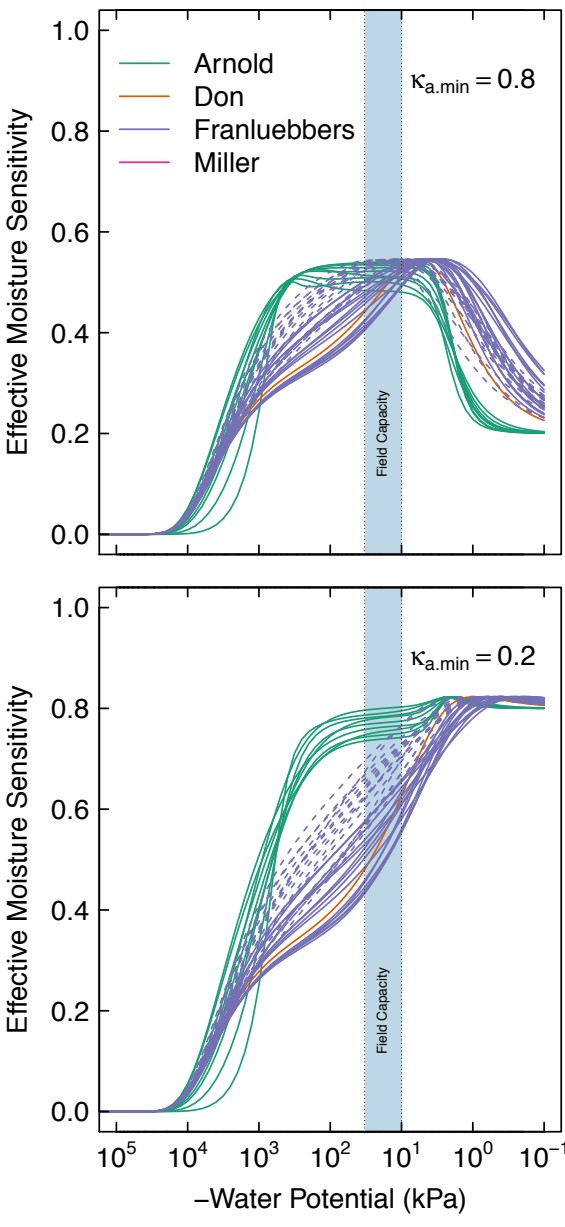

Figure 5: Effective soil moisture sensitivity functions for all the soils. These curves were calculated as illustrated in Figure 2.





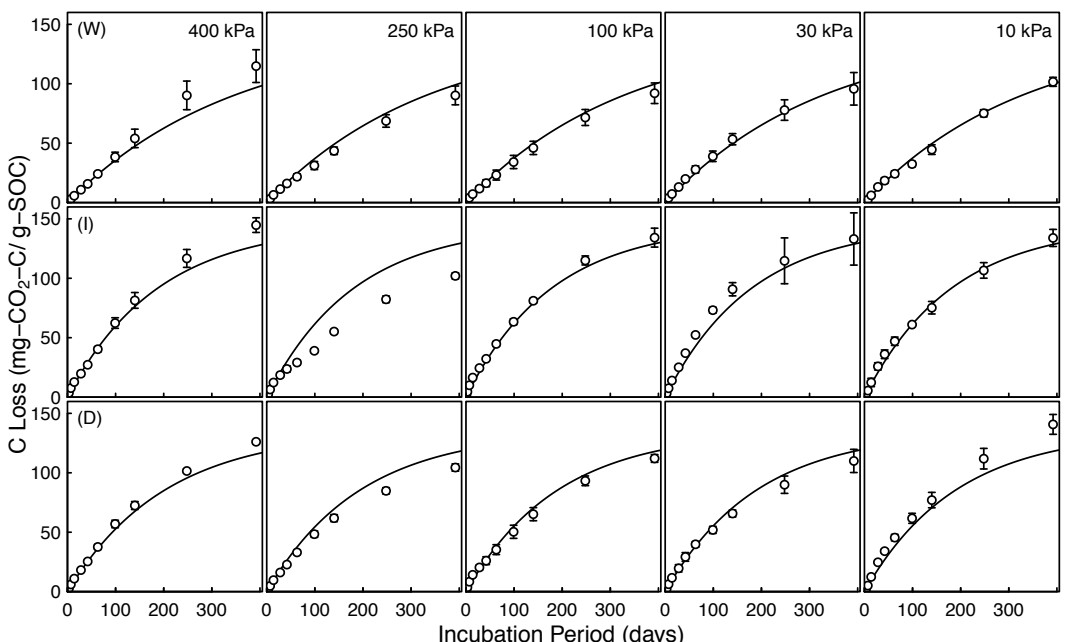

Figure 6: Evolution of $CO_2$ during 395 day incubation of soils collected from Dana Meadows (Yosemite National Park) from 0-10 cm depth over a wide range of water potentials. Other depths are provided in supplemental data. Soils from three hydrologic regimes (wet, intermediate, and dry) are shown.

Figure 7: Comparison of total SOC loss during 24 day incubation of 15 soils analyzed by Franzlue-bers (1999) (at naturally settled and compressed states); symbols are measured values and lines are model simulations using van Genuchten SWC curves and decomposition parameters, $C_0$ and $\kappa_\circ$, fitted to individually to each of the 15 soil types.





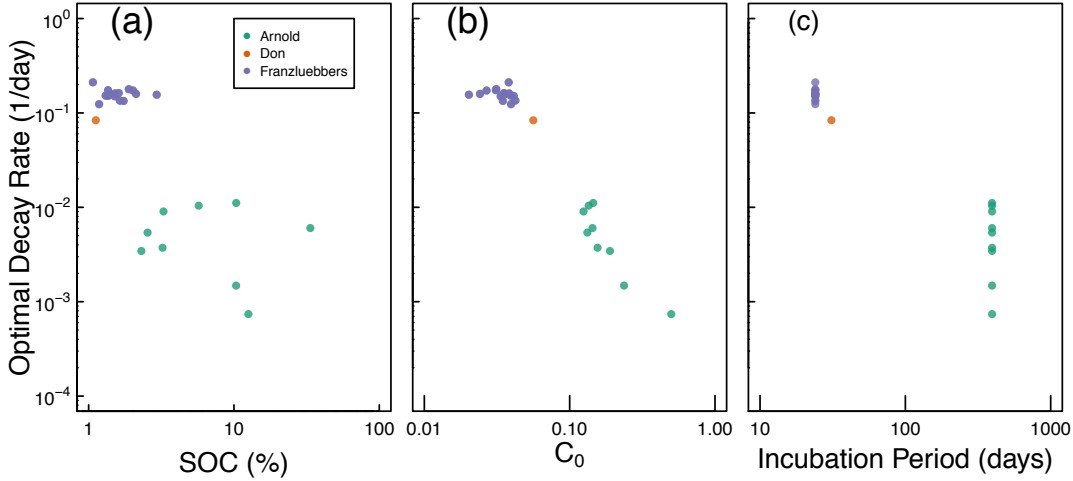

Figure 8: Relationships among fitted decomposition parameters $C_0$ and $\kappa_\circ$ as well as with soil organic C content (SOC) and length of incubation period.

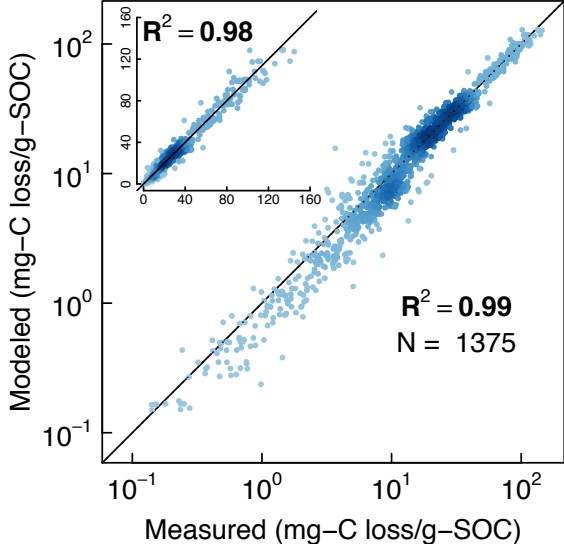

Figure 9: Comparison of model simulations with measured cumulative $CO_2$ evolution data from all incubation studies at steady-water content.





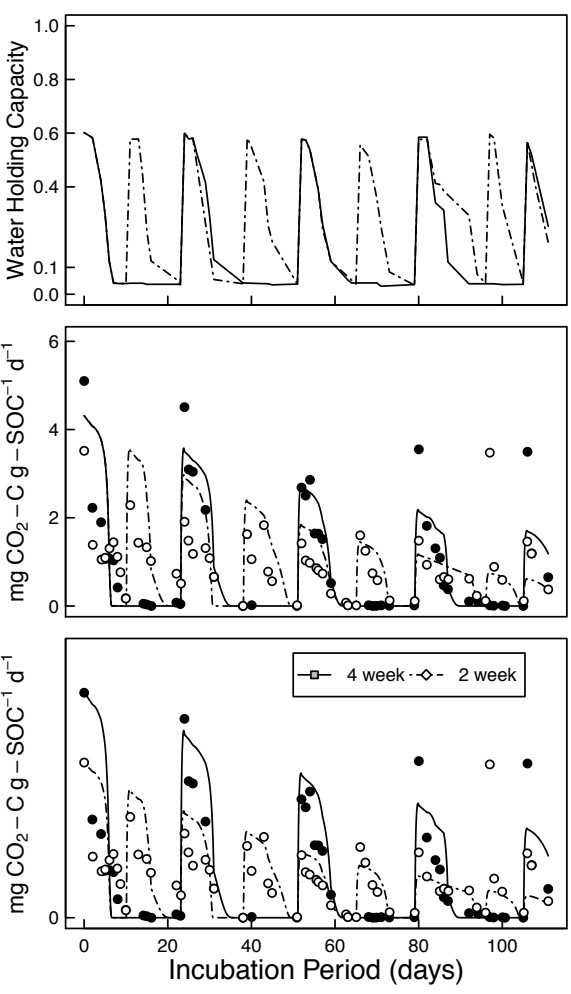

Figure 10: Comparison of measured $CO_2$ efflux during 2- and 4-week rewetting experiment (a) with model predicted efflux assuming (b) identical decomposition parameters for both wetting intervals and (c) separate decomposition parameters for the two wetting intervals.





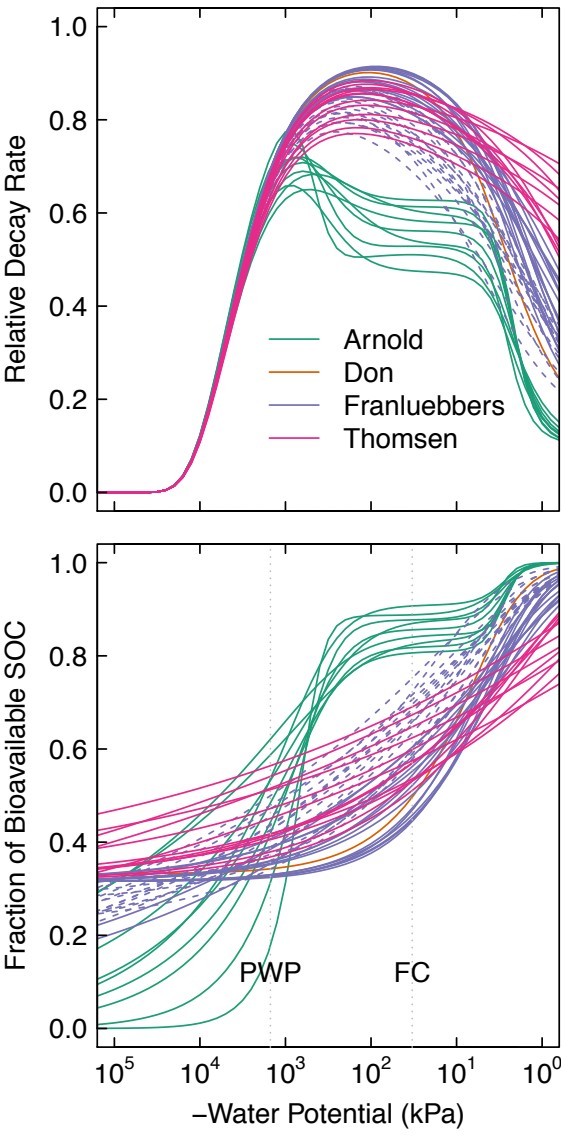

Figure A.1: Moisture-dependent, relative (dimensionless) parameters of 51 different soils: (a) decay rate and (b) fraction of bioavailable SOC. Each of the curves are entirely dependent only on the water retention characteristic of the respective soils

.





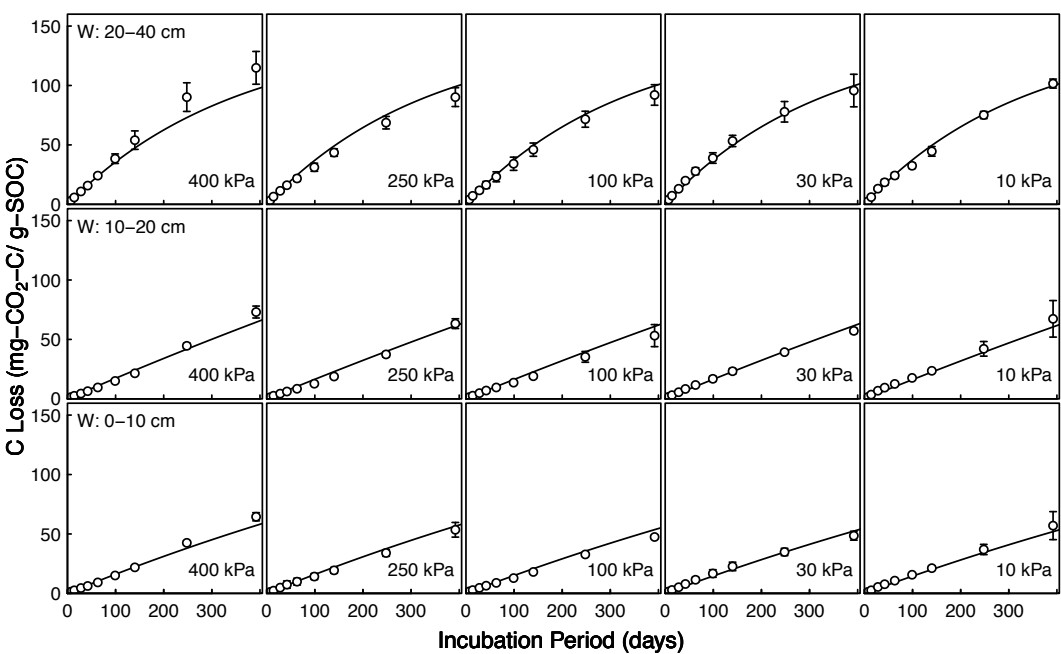

Figure A.2: (part 1/3) Decomposition experiments of Arnold et al. fitted $CO_2$ evolution data from 395-day incubation experiment: Part 1 wet meadow.



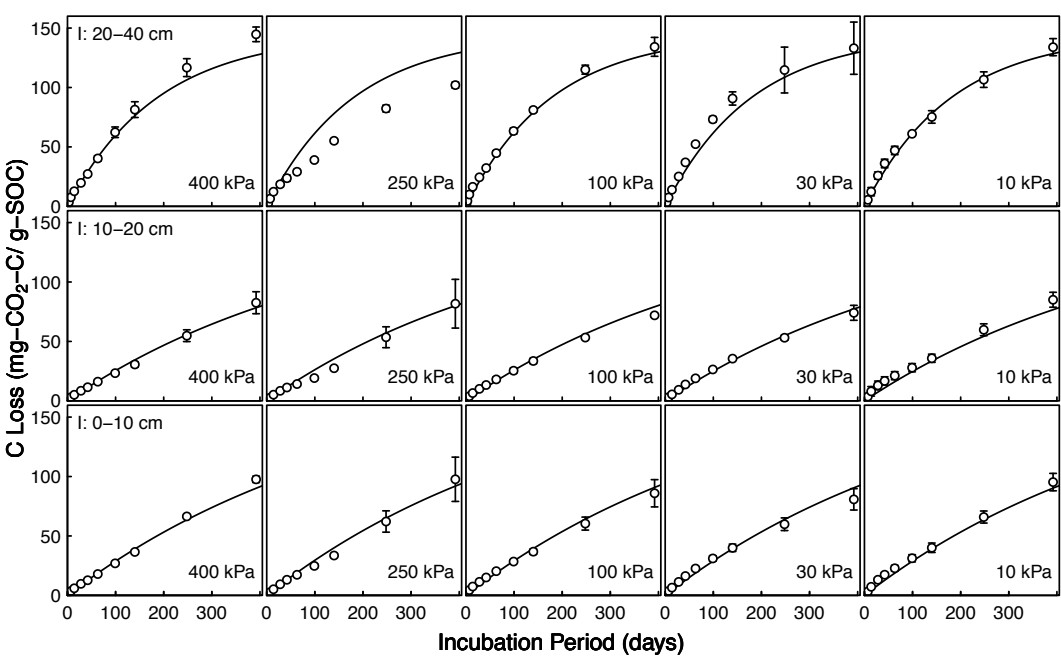

Figure A.2: (part 2/3) Decomposition experiments of Arnold et al. fitted $CO_2$ evolution data from 395-day incubation experiment: Part 2 intermediate meadow.





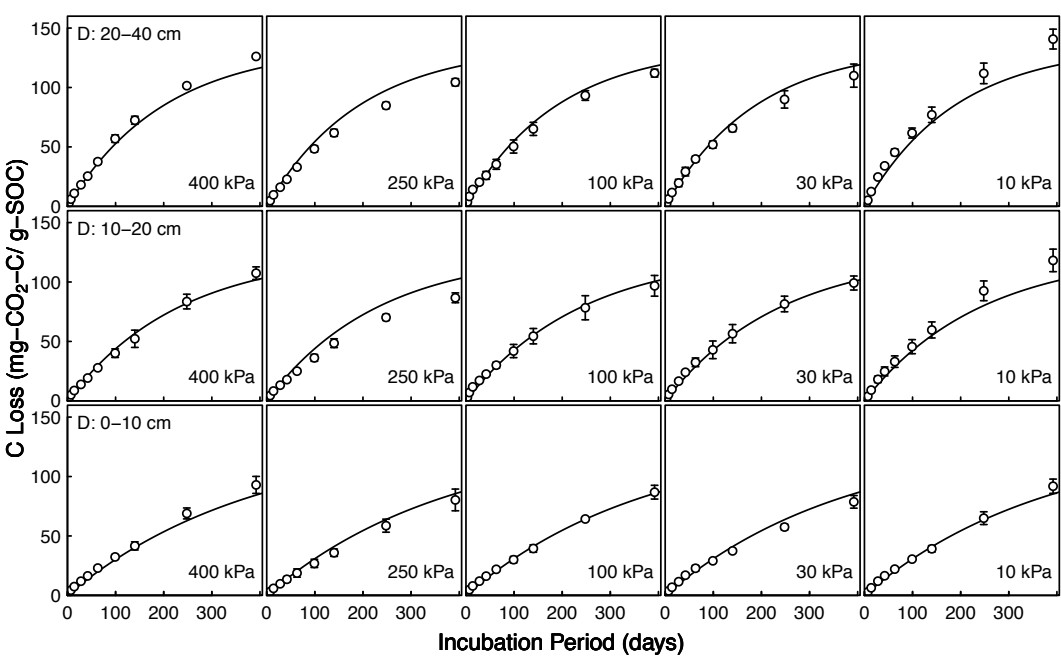

Figure A.2: (part 3/3) Decomposition experiments of Arnold et al. fitted $CO_2$ evolution data from 395-day incubation experiment: Part 3 dry meadow.



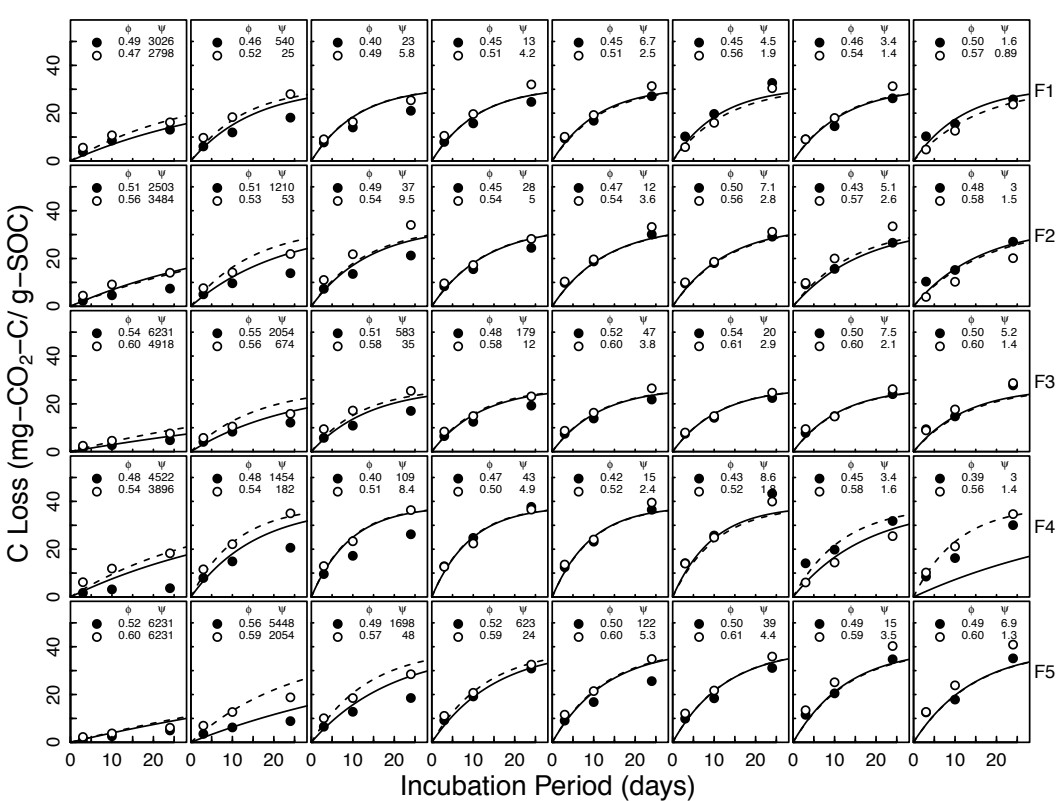

Figure A.3: (part 1/3) Decomposition experiments of Franzluebbers et al; fitted CO2 evolution data. Fifteen different soils packed at two bulkd density values incubated eight matric potential levels for 24 days. The porosity, water potential and RMSE of each sample are shown inside



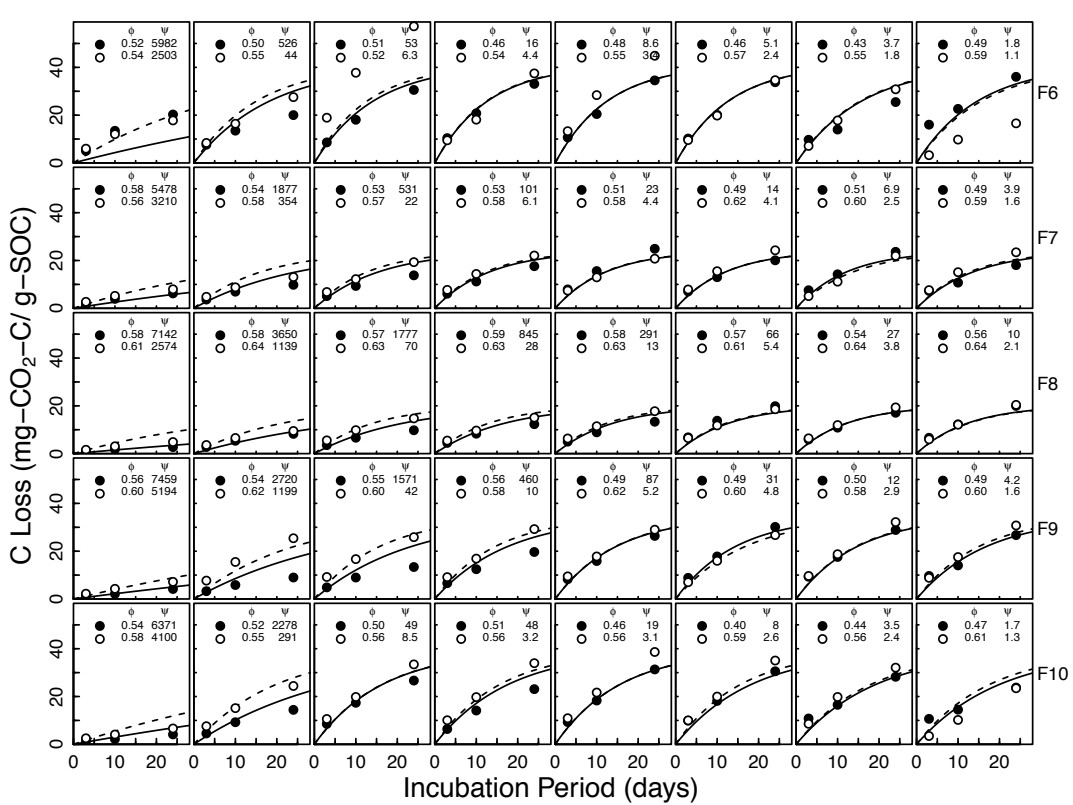

Figure A.3: (part 2/3) Decomposition experiments of Franzluebbers et al; fitted CO2 evolution data. Fifteen different soils packed at two bulkd density values incubated eight matric potential levels for 24 days. The porosity, water potential and RMSE of each sample are shown inside





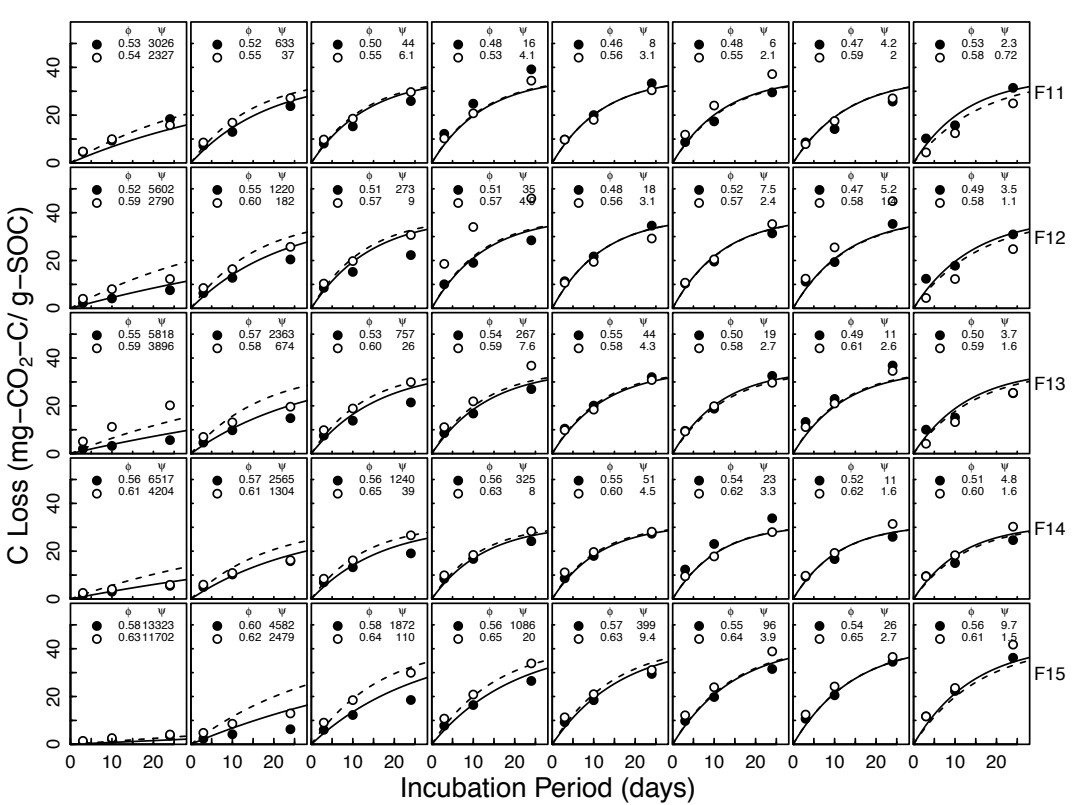

Figure A.3: (part 3/3) Decomposition experiments of Franzluebbers et al; fitted CO2 evolution data. Fifteen different soils packed at two bulkd density values incubated eight matric potential levels for 24 days. The porosity, water potential and RMSE of each sample are shown inside





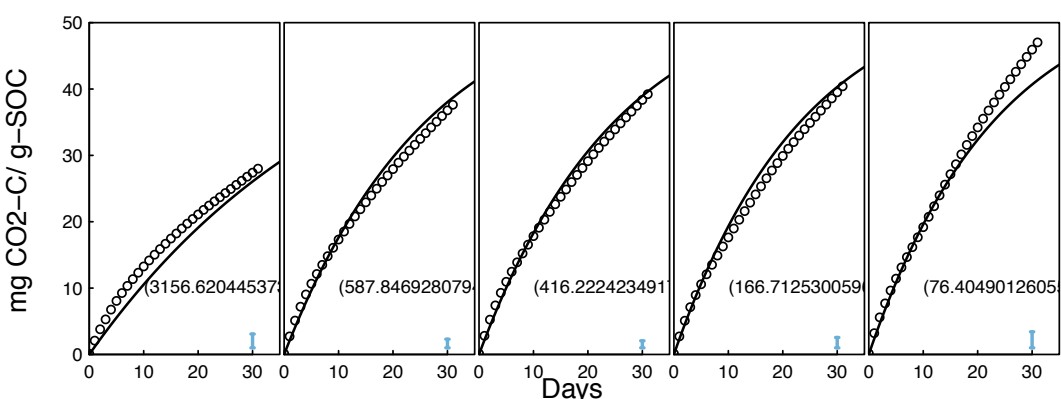

Figure A.4: Decomposition experiments of Don (data from Moyano); fitted CO2 evolution data. Error bars denote RMSE. Soils from three hydrologic regimes and three depths incubated at five matric potentials for 400 days.



**Table A1.** Best Water Retention Curve and SOM dynamics model parameters

| Soil Source and ID | Texture | | | SOC | $\kappa_0$ | $C_0$ | $\theta_r$ | $\theta_s$ | van Genuchten | | Durner | | | | |
|---|---|---|---|---|---|---|---|---|---|---|---|---|---|---|---|
| | sand | silt | clay | SOC | $\kappa_0$ | $C_0$ | $\theta_r$ | $\theta_s$ | $\alpha$ | $n$ | $\alpha_1$ | $n_1$ | $\alpha_2$ | $n_2$ | $w$ |
| | sand | silt | clay | [g/g] | [yr$^{-1}$] | $C_0$ | [—] | [—] | [yr$^{-1}$] | [—] | $\alpha_1$ | [—] | $\alpha_2$ | [—] | [—] |
| Arnold D.B | 0.650 | 0.260 | 0.090 | 0.025 | 0.005 | 0.132 | NA | NA | NA | NA | 0.004 | 1.330 | 0.413 | 5.000 | 0.282 |
| Arnold D.M | 0.650 | 0.270 | 0.060 | 0.033 | 0.009 | 0.125 | NA | NA | NA | NA | 0.002 | 2.556 | 0.397 | 3.194 | 0.231 |
| Arnold D.T | 0.670 | 0.280 | 0.050 | 0.057 | 0.010 | 0.135 | NA | NA | NA | NA | 0.003 | 1.660 | 0.376 | 4.557 | 0.344 |
| Arnold I.B | 0.610 | 0.320 | 0.070 | 0.023 | 0.003 | 0.189 | NA | NA | NA | NA | 0.003 | 1.974 | 0.395 | 3.684 | 0.265 |
| Arnold I.M | 0.640 | 0.310 | 0.050 | 0.032 | 0.004 | 0.156 | NA | NA | NA | NA | 0.005 | 1.490 | 0.338 | 4.629 | 0.227 |
| Arnold I.T | 0.710 | 0.230 | 0.060 | 0.104 | 0.011 | 0.145 | NA | NA | NA | NA | 0.007 | 1.301 | 0.386 | 4.176 | 0.300 |
| Arnold W.B | 0.730 | 0.250 | 0.050 | 0.103 | 0.001 | 0.237 | NA | NA | NA | NA | 0.003 | 1.753 | 0.391 | 3.545 | 0.173 |
| Arnold W.M | 0.640 | 0.320 | 0.040 | 0.126 | 0.001 | 0.500 | NA | NA | NA | NA | 0.002 | 3.896 | 0.404 | 2.735 | 0.214 |
| Arnold W.T | NA | NA | NA | 0.335 | 0.006 | 0.144 | NA | NA | NA | NA | 0.003 | 1.800 | 0.381 | 4.056 | 0.327 |
| Don NA | 0.807 | 0.103 | 0.090 | 0.011 | 0.146 | 0.064 | 0.050 | 0.407 | 0.351 | 1.763 | NA | NA | NA | NA | NA |
| Franz. Comp. F_1 | 0.820 | 0.090 | 0.090 | 0.014 | 0.174 | 0.031 | 0.036 | 0.458 | 0.616 | 1.398 | NA | NA | NA | NA | NA |
| Franz. Comp. F_2 | 0.760 | 0.120 | 0.120 | 0.015 | 0.150 | 0.034 | 0.048 | 0.480 | 0.315 | 1.523 | NA | NA | NA | NA | NA |
| Franz. Comp. F_3 | 0.660 | 0.165 | 0.175 | 0.020 | 0.173 | 0.027 | 0.000 | 0.518 | 0.726 | 1.236 | NA | NA | NA | NA | NA |
| Franz. Comp. F_4 | 0.710 | 0.100 | 0.190 | 0.011 | 0.211 | 0.038 | 0.020 | 0.439 | 0.226 | 1.336 | NA | NA | NA | NA | NA |
| Franz. Comp. F_5 | 0.570 | 0.170 | 0.260 | 0.013 | 0.153 | 0.039 | 0.000 | 0.509 | 0.409 | 1.216 | NA | NA | NA | NA | NA |
| Franz. Comp. F_6 | 0.775 | 0.125 | 0.100 | 0.014 | 0.152 | 0.041 | 0.035 | 0.480 | 0.565 | 1.415 | NA | NA | NA | NA | NA |
| Franz. Comp. F_7 | 0.670 | 0.170 | 0.160 | 0.021 | 0.159 | 0.024 | 0.000 | 0.522 | 0.855 | 1.249 | NA | NA | NA | NA | NA |
| Franz. Comp. F_8 | 0.510 | 0.275 | 0.215 | 0.029 | 0.156 | 0.020 | 0.000 | 0.570 | 0.770 | 1.223 | NA | NA | NA | NA | NA |
| Franz. Comp. F_9 | 0.540 | 0.205 | 0.255 | 0.017 | 0.134 | 0.035 | 0.000 | 0.522 | 0.612 | 1.221 | NA | NA | NA | NA | NA |
| Franz. Comp. F_10 | 0.610 | 0.145 | 0.245 | 0.012 | 0.124 | 0.039 | 0.017 | 0.479 | 0.486 | 1.320 | NA | NA | NA | NA | NA |
| Franz. Comp. F_11 | 0.780 | 0.110 | 0.110 | 0.014 | 0.158 | 0.036 | 0.024 | 0.496 | 0.743 | 1.375 | NA | NA | NA | NA | NA |
| Franz. Comp. F_12 | 0.725 | 0.125 | 0.150 | 0.015 | 0.161 | 0.038 | 0.018 | 0.506 | 0.698 | 1.307 | NA | NA | NA | NA | NA |
| Franz. Comp. F_13 | 0.615 | 0.175 | 0.210 | 0.016 | 0.163 | 0.035 | 0.000 | 0.528 | 0.808 | 1.234 | NA | NA | NA | NA | NA |
| Franz. Comp. F_14 | 0.535 | 0.220 | 0.245 | 0.019 | 0.179 | 0.031 | 0.000 | 0.547 | 1.015 | 1.233 | NA | NA | NA | NA | NA |
| Franz. Comp. F_15 | 0.490 | 0.180 | 0.330 | 0.016 | 0.135 | 0.042 | 0.000 | 0.567 | 0.912 | 1.204 | NA | NA | NA | NA | NA |
| Franz. Nat. F_1 | 0.820 | 0.090 | 0.090 | 0.014 | 0.174 | 0.031 | 0.052 | 0.458 | 0.527 | 1.694 | NA | NA | NA | NA | NA |
| Franz. Nat. F_2 | 0.760 | 0.120 | 0.120 | 0.015 | 0.150 | 0.034 | 0.055 | 0.480 | 0.435 | 1.756 | NA | NA | NA | NA | NA |



| Soil Source and ID | Texture | | | SOC | $\kappa_0$ | $c_0$ | $\theta_r$ | $\theta_s$ | van Genuchten | | Durner | | | | |
| | sand | silt | clay | | | | | | $\alpha$ | $n$ | $\alpha_1$ | $n_1$ | $\alpha_2$ | $n_2$ | $w$ |
| | sand | silt | clay | [g/g] | [yr$^{-1}$] | $c_0$ | [–] | [–] | yr$^{-1}$ | [–] | $\alpha_1$ | [–] | $\alpha_2$ | [–] | [–] |
| Franz. Nat. F_3 | 0.660 | 0.165 | 0.175 | 0.020 | 0.173 | 0.027 | 0.059 | 0.518 | 1.365 | 1.374 | NA | NA | NA | NA | NA |
| Franz. Nat. F_4 | 0.710 | 0.100 | 0.190 | 0.011 | 0.211 | 0.038 | 0.053 | 0.439 | 0.623 | 1.504 | NA | NA | NA | NA | NA |
| Franz. Nat. F_5 | 0.570 | 0.170 | 0.260 | 0.013 | 0.153 | 0.039 | 0.046 | 0.509 | 1.318 | 1.310 | NA | NA | NA | NA | NA |
| Franz. Nat. F_6 | 0.775 | 0.125 | 0.100 | 0.014 | 0.152 | 0.041 | 0.055 | 0.480 | 0.536 | 1.772 | NA | NA | NA | NA | NA |
| Franz. Nat. F_7 | 0.670 | 0.170 | 0.160 | 0.021 | 0.159 | 0.024 | 0.059 | 0.522 | 0.834 | 1.479 | NA | NA | NA | NA | NA |
| Franz. Nat. F_8 | 0.510 | 0.275 | 0.215 | 0.029 | 0.156 | 0.020 | 0.000 | 0.570 | 1.843 | 1.262 | NA | NA | NA | NA | NA |
| Franz. Nat. F_9 | 0.540 | 0.205 | 0.255 | 0.017 | 0.134 | 0.035 | 0.000 | 0.522 | 2.141 | 1.237 | NA | NA | NA | NA | NA |
| Franz. Nat. F_10 | 0.610 | 0.145 | 0.245 | 0.012 | 0.124 | 0.039 | 0.057 | 0.479 | 0.664 | 1.701 | NA | NA | NA | NA | NA |
| Franz. Nat. F_11 | 0.780 | 0.110 | 0.110 | 0.014 | 0.158 | 0.036 | 0.056 | 0.496 | 0.662 | 1.709 | NA | NA | NA | NA | NA |
| Franz. Nat. F_12 | 0.725 | 0.125 | 0.150 | 0.015 | 0.161 | 0.038 | 0.058 | 0.506 | 0.751 | 1.655 | NA | NA | NA | NA | NA |
| Franz. Nat. F_13 | 0.615 | 0.175 | 0.210 | 0.016 | 0.163 | 0.035 | 0.059 | 0.528 | 0.829 | 1.499 | NA | NA | NA | NA | NA |
| Franz. Nat. F_14 | 0.535 | 0.220 | 0.245 | 0.019 | 0.179 | 0.031 | 0.062 | 0.547 | 1.198 | 1.485 | NA | NA | NA | NA | NA |
| Franz. Nat. F_15 | 0.490 | 0.180 | 0.330 | 0.016 | 0.135 | 0.042 | 0.063 | 0.567 | 1.886 | 1.358 | NA | NA | NA | NA | NA |