# Peer review of "On the role of soil water retention characteristic on aerobic microbial respiration"

_Biogeosciences, 2018_

## Referee Comment (RC1) · Anonymous Referee #1 · 7 Jul 2018

This manuscript proposes a mathematical model to account for the effects of water in the decomposition of soil organic matter. This is an important study, because it helps to disentangle the debate on whether soil water content or soil water potential is the most appropriate measure to account for the effects of water on soil microbial activity (Sierra et al., 2015; Vicca et al., 2012). The authors address this issue by proposing the use of soil water retention curves, which combine both metrics and have a very well stablished theory in soil physics and hydrology. To test the model, the authors use a number of datasets from published studies and show that the approach works well at fitting data from incubation experiments.

The idea presented in this study has the potential to have impact in the way the effects of soil water are accounted for in different ecological studies. However, there are problems in the presentation of the approach that need to be addressed before acceptance for publication.

**1  Major comments**

I have problems following the model description (mostly eqs. 11 to 14) because problems of mathematical notation and apparent misuse of assumptions. I will explain this on a point by point basis.

- Page 11, line 4. The authors state that the model of equation 5 can be solved under arbitrary fluctuations of soil water status, i.e. $\theta(t)$ and $\psi(t)$. However, this is not possible to do in close form unless you have a very specific function that shows how $\theta$ and $\psi$ change over time; and if you have these functions, it is very unlikely that you will obtain an analytical solution. I would say that this assumption is wrongly stated here, and the authors should acknowledge that the analytical expressions they provide only apply for constant soil water status. Later on page 16, lines 16-17, the authors correctly point out that only numerical solutions are possible for the time-dependent case. This is obviously contradictory to what is stated on page 11.

- The upper limit of integration in equation 11 is with respect to time, but $K(\theta, \psi)$ is not time-dependent as expressed in equation 12. It seems to me that you may want to integrate over $\theta$ or $\psi$, but not $t$.

- The solution of equation 5 is $C(t) = C_0 \exp(-k \cdot t)$. I assume that your intention is to be able to replace equations 7 to 10 for $k$ as expressed in equation 6. If so, then equation 11 is missing a minus sign and $t$.

- Why do you need $C_0$ in equation 12? I cannot trace it back from the previous equations. Also, what happened to $\lambda$? Shouldn't it go here?

- Equation 14 doesn't seem right to me. What you probably want is to compute the integral of the respired carbon, i.e. $C_{CO_2}(t) = \int_0^t R(t)dt = \int_0^t 1 - C(t)dt.$

These problems in the mathematical description are important, because you cannot propose a new model if you cannot describe it correctly. Since the authors do not provide the code used for their computations it is also impossible to test whether the mathematical description corresponds with the implementation.

Another limitation I see in this study is the lack of contrast with a related model that may perform poorly with respect to the newly proposed model. To my knowledge, the only model that can also deal with these multiple limitations is the DAMM model of Davidson et al. (2014). It would be very helpful if the new model is contrasted against DAMM or other model to more explicitly see the advantage of the new method.

It is my impression that this model requires the availability of water retention curves for its use. This obviously implies an extra effort in terms of data collection. Can the authors elaborate more on this potential limitation of the method?

**2 Minor comments**

- Page 6, lines 9-10. Which one is eq. 2 and 3 in fig. 1, i.e. red or blue?

- Page 6, line 16. Remove point.

- Equation 7. Here it may be good to remind the reader that matric potential is negative, and therefore $k$ can't be higher than 1.

- Page 14, line 8. Can you provide a justification or a reference for this choice of parameter value?

- Fig 7. What is the difference between the red and the black lines?

**References**

Davidson, E. A., Savage, K. E., and Finzi, A. C. (2014). A big-microsite framework for soil carbon modeling. *Global Change Biology*, 20(12):3610–3620.

Sierra, C. A., Trumbore, S. E., Davidson, E. A., Vicca, S., and Janssens, I. (2015). Sensitivity of decomposition rates of soil organic matter with respect to simultaneous changes in temperature and moisture. *Journal of Advances in Modeling Earth Systems*, 7(1):335–356.

Vicca, S., Gilgen, A. K., Camino Serrano, M., Dreesen, F. E., Dukes, J. S., Estiarte, M., Gray, S. B., Guidolotti, G., Hoeppner, S. S., Leakey, A. D. B., Ogaya, R., Ort, D. R., Ostrogovic, M. Z., Rambal, S., Sardans, J., Schmitt, M., Siebers, M., van der Linden, L., van Straaten, O., and Granier, A. (2012). Urgent need for a common metric to make precipitation manipulation experiments comparable. *New Phytologist*, 195(3):518–522.

---

## Referee Comment (RC2) · Anonymous Referee #2 · 11 Jul 2018

General comments

The contribution by Ghezzehei and co-authors presents a model describing the responses of microbial respiration to changes in soil water. The proposed approach follows the work by Skopp et al. (1990; cited in the Discussion paper) and defines a set of limiting functions that affect respiration: one for oxygen availability, one for aqueous diffusivity, and one for matric potential effects (microbial activity limitation). The latter function represents an improvement over the original model by Skopp et al., but it is analogous to some other recent papers (see details below). These limiting functions are then combined in a factor that rescales the first order decay constant regulating carbon release from a single pool of organic carbon. This model is then parameterized using measured respiration-soil water relations for a number of soils. This topic

is timely given the uncertainties in modelling respiration-soil water relations, and suitable for Biogeosciences; however, I have some concerns regarding the novelty of the proposed approach, I found the model description at times confusing, and there are several inconsistencies and language/presentation issues.

Specific comments

As acknowledged by the authors, the use of combined gas and aqueous diffusion limiting functions to predict respiration-soil water relations had been proposed by Skopp et al. (1990) and used in many occasions later. The matric potential-dependent function capturing reductions in microbial activity is a more novel addition, but similar functions have been recently proposed and used to capture respiration-soil water trends observed in laboratory studies (Yan et al. 2016; Manzoni et al. 2016). It might also be worth looking at other recent papers (some not available at the time this contribution was submitted) using a comparable approach, though with equations derived in different ways (Tang and Riley, 2013; Yan et al. 2018; Moyano et al. 2018). Considering these previous papers, some statements in the Discussion and Conclusions section seem to overstate the novelty of this contribution (P18, L4-5; P19, L9).

The model description is not always clear and there are several inconsistencies in the way parameters are defined. For example, in Eq. 10, the aqueous diffusivity $D\_W$ does not have the dimensions of a diffusivity ($L^2/T$), but is non-dimensional. The symbol $C\_A$ in the same equation is not used elsewhere. In Eq. 11-13, which are used to fit the data, $C\_A$ does not appear, so 'accessibility' does not play a role, unless $C\_0$ is interpreted as the 'accessible' organic carbon (but that is defined as 'initial active carbon'). Moreover, the units in Eq. 11-12 do not match up: with $K$ defined as in Eq. 12, the exponent in Eq. 11 is not non-dimensional, but has the same units of $C\_0$. Towards the end of the manuscript, a "curve lambda" is mentioned (P19, L3), but lambda is only used as a parameter before. Overall, these issues make the reading and interpretation of results difficult.

Some choices of the soil moisture characteristic curves appear arbitrary. How were unimodal vs. bimodal curves selected? At the dry end of the soil moisture characteristic curves in Fig. 4, for example, there appear to be a sharp decrease in water content – possibly a sign that a bimodal curve could work better? I would suggest selecting curves using a more objective criterion based on goodness of fit and robustness (e.g., AIC).

Minor comments

- Please check the whole text for grammar mistakes and inconsistent formatting of citations (e.g., author names in capital, erroneous use of brackets); some of these issues are highlighted below P1, L17: "comparing" P1, L22: "Yuste" P3, L6: "nitrification rate… correlates" P6, L1: if alpha refers to matric potential at maximum drainage, I am not sure I understand why D_0 (a function of alpha) refers to the modal rather than maximum pore throat diameter P6, L9: "top axis of the figure" – which figure? I would refer to the figure number P6, L15: "unimodal" P6, L16: extra full stop? This sentence appears incomplete P7, L12: check use of brackets - "Chowdhury et al. (2011b)" P7, L16: "Watson" P8, L4: this sentence appears incomplete P11, L17: "important to note" P14, L8: but in Figure 5, k_a,min=0.8 as well P15, L4: what does "explained in its entirety" mean? Based on which performance metric? P15, L17: "soils that were…" P16: to avoid having incubation duration as a confounding factor, only the first data points from the Arnold et al. (2015) study could be used P16, L21: more than inter-sample differences, the data from Miller et al. (2005) show strong Birch effect (Birch 1958) – longer dry periods trigger larger respiration pulses. This effect, which is widespread, cannot be captured by the proposed model P17, L15: delete "in the" P17-18: the structure of the Discussion and Conclusion section is a bit strange, with two introductory paragraphs and a single numbered subsection P26, last line of the caption: "diameter" Figure 2: check if labels (B) and (C) are correctly placed; the caption is not consistent with the figure and does not explain what panel (d) shows P30, caption: no explanation of the difference between top and bottom panel is provided Figure 6: check panel

labels – now only (W), (I), and (D) appear as labels Figure 7: not clear what is the difference between red and black curves Figure A3: "bulk density" Figure A4: what are the numbers in brackets? Is the number of significant digits reasonable?

References

Birch, H. F. 1958. The effect of soil drying on humus decomposition and nitrogen availability Plant and Soil 10:9-31. Manzoni, S., F. Moyano, T. Kätterer, and J. Schimel. 2016. Modeling coupled enzymatic and solute transport controls on decomposition in drying soils. Soil Biology and Biochemistry 95:275-287. Moyano, F. E., Vasilyeva, N., and Menichetti, L.: Diffusion based modelling of temperature and moisture interactive effects on carbon fluxes of mineral soils, Biogeosciences Discuss., https://doi.org/10.5194/bg-2018-95, in review, 2018. Tang, J. Y., and W. J. Riley. 2013. A total quasi-steady-state formulation of substrate uptake kinetics in complex networks and an example application to microbial litter decomposition. Biogeosciences 10:8329-8351. Yan, Z., Liu, C., Todd-Brown, K.E. et al. 2016. Pore-scale investigation on the response of heterotrophic respiration to moisture conditions in heterogeneous soils. Biogeochemistry 131: 121–134, https://doi.org/10.1007/s10533-016-0270-0 Yan, Z., B. Bond-Lamberty, K. E. Todd-Brown, V. L. Bailey, S. Li, C. Liu, and C. Liu. 2018. A moisture function of soil heterotrophic respiration that incorporates microscale processes. Nature communications 9:2562.

---

## Referee Comment (RC3) · A. Ebrahimi (Referee) · 16 Jul 2018

The manuscript proposes a new modeling framework that integrates the important role of soil water potential on regulating the rate of soil respiration. The model is built on assuming a single pool soil organic matter (SOM) where a first-order kinetics for the rate of SOM decomposition is considered. Authors have expanded the decay rate of SOM ($k$ parameter) to incorporate for the role of biophysical factors, mainly matric potential. This step is performed by a simple and testable exponential relationship between the decay rate and matric potential. The model is then expanded to include variations in oxygen and substrate diffusion as a function of matric potential and soil depth. The simple nature of proposed mathematical framework allows its application for large-scale carbon cycle and climate models while preserving the effects of some

of the key biophysical factors. This step is performed nicely in this model by reducing the number of calibration parameters and limiting them to some measurable quantities. The model is ultimately tested again good amount of datasets.

Overall, the technical quality of the manuscript is high and the proposed model has potential to be used in other biogeochemical gas flux models to account for the role of water content and potential, individually. I have some minor comments and recommendations that I believe could help the manuscript to be stronger and accessible for broader audiences.

My main suggestion is to better discuss uncertainties and limitations associated with the previously developed models that the current model aims to address those limitations. At the moment, it is not completely clear how incorporating matric potential into the model improves the model predictions compared to the models without this feature.

While the idea of using SWC is nice, the implementation and formulation is rather confusing and hard to follow. The main problem might be the inadequate description of the parameters and the links of parameters through the equations.

I also found that the manuscript is a bit bulky in the introduction and method descriptions. I suggest shortening the introduction and methods. While some of the discussions and examples in the introduction and method are informative, I think it might be destructing. For instance examples and discussions on nitrification process could be misleading, since the main story is about respiration and the connection between respiration and nitrification processes is not immediately clear even though both could be aerobic processes. If this part is necessary, I would suggest to provide a discussion on its need.

My other suggestion is to better explain the difference between water content and matric potential, maybe in a schematic. For instance the independent relationship of water potential from water content and its effects on osmotic potential that is discussed in the manuscript is not so clear. This is important motivation of the paper and could be illustrated a little bit more. Meanwhile, the effects of osmotic potential are discussed in the introduction, but its incorporation in the model is not so clear, even though it has been assumed that Eq. 6 could also account for osmotic potential.

Minor comments: Page 1, Line 21: "are strongly correlated heterotrophic respiration rates" grammar error? References are not consistent. Some author names are capital and some are not. Page 2, line 3: "films is dependent" grammar? "Moisture sensitivity curve" is probably not accurate terminology. I suggest to define moisture sensitivity term. In page 4 line 18, "biophysical rates". Here it is not clear what authors mean. In Eq. 6 and 7 different k parameters are used. I would suggest to better define these parameters. The current version is a bit confusing. Section 2.2 "SOM dynamics modeling" is very long that makes it hard to read and follow the method. I suggest breaking down this section into subsections with detailed subheadings. In line 6, page 8, I think the difference between gas and liquid diffusion coefficients of oxygen is about 4 orders of magnitude, I suggest checking the number, once more. Figure A1 is unclear. At the moment, it is unclear what dashed lines mean. PWP and FC could be defined in the caption of the figure. "Bioavailable SOC" should be defined. The term has not been defined and discussed in the rest of the manuscript.

---

## Author Comment (AC1) · 17 Oct 2018

**Major Comments**

**Comment 1:** Page 11, line 4. The authors state that the model of equation 5 can be solved under arbitrary fluctuations of soil water status, i.e. $\theta(t)$ and $\psi(t)$. However, this is not possible to do in close form unless you have a very specific function that shows how $\theta$ and $\psi$ change over time; and if you have these functions, it is very unlikely that you will obtain an analytical solution. I would say that this assumption is wrongly stated here, and the authors should acknowledge that the analytical expressions they provide only apply for constant soil water status. Later on page 16, lines 16-17, the authors correctly point out that only numerical solutions are possible for the time-dependent

case. This is obviously contradictory to what is stated on page 11.

**Response 1:** We agree that a complete closed-form solution does not exist for arbitrary fluctuation. We left the integral as is in Eq 11 for this reason. We restated the above sentence as: "The SOM dynamics under arbitrary fluctuation of soil water status (i.e., $\theta(t)$ and $\psi(t)$) can be described by re-arranging Eq. (5), subject to initial active pool of SOC $C(t=0) = C_0$, as..". Also we added the following sentence right after the equation. "Note that closed form solution for the integral in Eq. 11 exists only at steady water content and water potential status..."

**Comment 2:** The upper limit of integration in equation 11 is with respect to time, but $K(\theta, \psi)$ is not time-dependent as expressed in equation 12. It seems to me that you may want to integrate over $\theta$ or $\psi$, but not t.

**Response 2:** The dependence of water content and matric potential was stated in line 4 right above Eq 11, therefore it was implied in equations 11 and 12. We now explicitly show this dependence in Eq 11:

$$C(t) = C_0 \exp\left(-\kappa_\circ \int_0^t K[\theta(t), \psi(t)]d\tau\right)$$

Equation 12 is an expression of instantaneous moisture sensitivity, therefore it is not necessary to express the dependence on time.

**Comment 3:** The solution of equation 5 is $C(t) = C_0 \exp(-kt)$. I assume that your intention is to be able to replace equations 7 to 10 for k as expressed in equation 6. If so, then equation 11 is missing a minus sign and t.

Thank you for pointing out this error. The missing negative sign to Eq 11 was added (see above correction).

**Comment 4:** Why do you need $C_0$ in equation 12? I cannot trace it back from the previous equations. Also, what happened to $\lambda$? Shouldn't it go here?

These were typographic errors. The effects of matric potential and accessibility (Eqs. 7 and 10) were inadvertently left out in Eq. 12, but have now been added.These typographic errors in the manuscript were not carried over to the the codes used for calculations. The corrected Eq 12 is:

$$K(\theta, \psi) = e^{\lambda\psi} \left\{ \kappa_{a,\min} + (1 - \kappa_{a,\min}) \left( \frac{\phi - \theta}{\phi} \right)^{1/2} \right\} \left( \frac{\theta}{\phi} \right)^{1/2}$$

**Comment 5:** Equation 14 doesn't seem right to me. What you probably want is to compute the integral of the respired carbon, i.e. $C_{CO_2} = \int_0^t R(t)d\tau = \int_0^t 1 - C(t)dt$.

We believe the integral in the suggested expression is redundant as $C(t)$ refers to the amount of SOC remaining at any given time (see Eq 11). Therefore, the respired C must be the difference between the initial and remaining C levels; i.e.

$$C_0 - C(t) = C_0 - C_0 \exp\left( \kappa_\circ \int_0^t K(\tau)d\tau \right) = C_0 \left\{ 1 - \exp\left( \kappa_\circ \int_0^t K(\tau)d\tau \right) \right\}$$

No change was made in response to this comment.

**Comment 6:** Another limitation I see in this study is the lack of contrast with a related model that may perform poorly with respect to the newly proposed model. To my

knowledge, the only model that can also deal with these multiple limitations is the DAMM model of Davidson et al. (2014). It would be very helpful if the new model is contrasted against DAMM or other model to more explicitly see the advantage of the new method.

> Although comparison with another model, such as DAMM, would provide interesting results the comparison would not address the key tenet of this manuscript–to accurately represent the role of soil structure as described by water retention characteristic. A more appropriate test for this model is to compare model performance against experiments in which the soil structure is manipulated such that contrasting water retention characteristics would be achieved for the same soil. Then comparing our model with other models that utilize only water content or matric potential would be meaningful. We hope that publishing this modeling framework would motivate researchers who may have data needed for such comparison to test the our hypothesis. No change was made in response to this comment.

**Comment 7:** It is my impression that this model requires the availability of water retention curves for its use. This obviously implies an extra effort in terms of data collection. Can the authors elaborate more on this potential limitation of the method?

> Yes, this is a limitation. It was made even more clear to us by the availability of only a handful datasets that we could use for testing our model, despite the fact that decomposition experiments at varying moisture statuses have been done numerous times. When WRC data is not available, a practical solution is to use pedo-transfer functions to determine the parameters of WRC. The following statement was added to the last section of the manuscript:

"Application of the proposed model requires availability of water retention characteristic, which may pose practical limitation in cases when water retention data cannot be readily acquired. Availability of only a handful datasets that we could use for testing the proposed model, despite the fact that decomposition experiments at varying moisture statuses have been done numerous times, is a clear evidence of this challenge. As a stopgap measure, it is possible to use pedotransfer functions to infer water retention parameters based on routinely measured soil characteristics such as texture, bulk density and organic matter content (Vereecken et al, 1989; Schaap et al, 2011; Van Looy et al, 2017)."

**Minor comments**

1. Page 6, lines 9-10. Which one is eq. 2 and 3 in fig. 1, i.e. red or blue?

We corrected it as:"In Fig 1, Eq (2) and (3) are illustrated by the solid blue line." We also added the following sentence after Eq 4: "In Fig 1, Eq (4) is illustrated by the solid red line. The corresponding bi-modal pore size density function is shown as red-shaded curve."

2. Page 6, line 16. Remove point.

Corrected

3. Equation 7. Here it may be good to remind the reader that matric potential is negative, and therefore k can't be higher than 1.

We dded: "Note that $\kappa_\psi \leq 1$ because matric potential cannot be positive ($\psi \leq 0$)."

4. Page 14, line 8. Can you provide a justification or a reference for this choice of parameter value?

The parameter denotes availability of oxygen under saturated moisture condition and ranges between 0 and 1. A value of 0 means complete lack of $O_2$ and a value of 1 means maximum $O_2$ concentration. For lab incubation samples, this value is expected to be dependent on sample size. In field conditions, soil depth is the most important factor that controls $a$. At this stage the parameter remains the most uncertain part of the proposed model and needs specific experiments to test and parameterize its value. For this paper, we chose the value of $\kappa_{a.\,\min} = 0.2$ based on the work of Ebrahimi and Or (Global Change Biology, 2016), which corresponds to the dissolved $[O_2]$ at moisture saturation of $\approx 0.9$.

5. Fig 7. What is the difference between the red and the black lines?

The red lines were inadvertently left. They represent a different approach that we tested earlier in the study. They have been removed.

---

## Author Comment (AC2) · 17 Oct 2018

**Major Comments**

**Comment 1:**As acknowledged by the authors, the use of combined gas and aqueous diffusion limiting functions to predict respiration-soil water relations had been proposed by Skopp et al. (1990) and used in many occasions later. The matric potential-dependent function capturing reductions in microbial activity is a more novel addition, but similar functions have been recently proposed and used to capture respiration-soil water trends observed in laboratory studies (Yan et al. 2016; Manzoni et al. 2016). It might also be worth looking at other recent papers (some not available at the time this contribution was submitted) using a comparable approach, though with equations

derived in different ways (Tang and Riley, 2013; Yan et al. 2018; Moyano et al. 2018). Considering these previous papers, some statements in the Discussion and Conclusions section seem to overstate the novelty of this contribution (P18, L4-5; P19, L9).

Thank you for directing us to the recent sources. We agree with the reviewer about some of the similarities with these sources and made changes accordingly. We added a sentence acknowledging that the diffusion limitation on substrate accessibility that we adopted in Eq. (10) is consistent with prior models (Tang and Riley, 2013; Yan et al. 2016; Manzoni et al. 2016). In the recent paper of Yan et al (2018), the effect of soil texture is captured by the empirical parameters that were fitted to the three soils. It is possible that the effect of SWC is implicitly contained within these tuned parameters as well. Our model was designed to directly address the physical effects of pore size distribution (as described by water retention curve). Other factors that are likely to depend on moisture status including enzyme activity and microbial community structure were not included. This was done to limit the number of tunable parameters and test to what extent water retention characteristic alone can explain moisture sensitivity. The complete moisture sensitivity function is given in Eq. 12 (typographic errors noted by this and the first reviewer have been corrected).

$$K(\theta, \psi) = e^{\lambda\psi} \left\{ \kappa_{a,\min} + (1 - \kappa_{a,\min}) \left( \frac{\phi - \theta}{\phi} \right)^{1/2} \right\} \left( \frac{\theta}{\phi} \right)^{1/2}$$

The differences in moisture sensitivity amongst all the soils considered in this study are shown in Figure 5. These Figures are comparable in pattern to those of Yan et al (2018; their Figure 5). The major difference being, in our model the shape of these curves is dependent only on the SWC parameters and $\kappa_{a,\min}$. The latter was kept consistent across all soils for simplicity

and because the available data was not adequate to test how this parameter varies with depth and/or sample size. In testing our model, the shape of the dimensionless moisture sensitivity curve was prescribed *a priori* based on independently acquired SWC parameters and fixed value of $\kappa_{a,\min} = 0.2$. Thus, the main contribution of our work, which is also a major departure from the models of Yan et al. (2018), Moyano et al. (2018) and their predecessors, is the absence of moisture-sensitivity parameters that are tuned to match with respiration data. This does not negate the importance of moisture dependence of enzymatic and/or microbial activities represented in these other models. To emphasize the above new contribution of this work, we added a new paragraph and a new figure in the discussion section showing the moisture sensitivity curves of the 12 US textural classes. SWC parameters for these soils were derived from Schaap et al (2001).

**Comment 2a:** The model description is not always clear and there are several inconsistencies in the way parameters are defined. For example, in Eq. 10, the aqueous diffusivity $D_W$ does not have the dimensions of a diffusivity ($L2/T$), but is non-dimensional. The symbol $C_A$ in the same equation is not used elsewhere. In Eq. 11-13, which are used to fit the data, $C_A$ does not appear, so 'accessibility' does not play a role, unless $C_0$ is interpreted as the 'accessible' organic carbon (but that is defined as 'initial active carbon'). Moreover, the units in Eq. 11-12 do not match up: with $K$ defined as in Eq. 12, the exponent in Eq. 11 is not non-dimensional, but has the same units of $C_0$. Towards the end of the manuscript, a "curve lambda" is mentioned (P19, L3), but lambda is only used as a parameter before. Overall, these issues make the reading and interpretation of results difficult.

As stated in the sentence preceding Eq. 10 accessibility "scales with the *relative* aqueous diffusivity", which implies that it is normalized by diffusivity of saturated soil. The expression in Eq. 10 is that of tortuosity, which by

definition is dimensionless. For clarity the sentences above Eq. 10 were
revised as follows:

"We assume the fraction of active SOC pool that is accessible to decomposers scales with relative aqueous diffusivity. Therefore, the accessible
fraction of the SOC pool is proportional to the liquid phase tortuosity. Here,
we use the Bruggeman expression for tortuosity,"

Eq. 12 had typographic errors that caused the confusion raised. The effects
of matric potential and accessibility (Eqs. 7 and 10) were inadvertently left
out in Eq. 12, but have now been added (see also response to Reviewer
**1). In addition, the equation represented a closed-form solution of the**
right-hand-side of Eq 11 for constant $\psi$ and $\theta$. These was typographic errors
in the manuscript but we verified that the codes we used for calculations
were correct. The corrected Eq 12 is given above.

**Comment 3:** Some choices of the soil moisture characteristic curves appear arbitrary.
How were unimodal vs. bimodal curves selected? At the dry end of the soil moisture
characteristic curves in Fig. 4, for example, there appear to be a sharp decrease
in water content – possibly a sign that a bimodal curve could work better? I would
suggest selecting curves using a more objective criterion based on goodness of fit and
robustness (e.g., AIC).

Multimodality of soil water retention curve can arise due to clear distinction between capillary and adsorptive forces. A fairly recent water retention
model by Peters (2013) and Iden and Durner (2013) (now known as Peters–Durner–Iden (PDI) model) suggests that most soils should exhibit
bimodality as the drying curve of adsorbed water usually exhibits different
pattern from that of water held by capillary forces. The transition from capillary dominated retention to adsorption dominated retention occurs at very

low matric potential levels ($< -100kPa$). But bimodality can also arise due to structure (e.g., aggregation or biopores) (Durner, 1994) that results in two (or more) distinct populations of pore sizes. The transition between macro-pore dominated retention and micro-pore dominated retention usually occurs at high matric potential. In this paper, the bimodal models were strictly used for soils that exhibit structure related bimodality. An alternative approach would have been to include adsorptive component to all the soils. This would mean that soils that also show additional structural effect need to be fitted with a trimodal model. None of the water retention data that we used have sufficient number of measurements to match the additional degrees of freedom that would be introduced by such model. Therefore we chose to use the classical van Genuchten unimodal model for all soils that exhibit bimodality at the dry end

**Minor comments** - Please check the whole text for grammar mistakes and inconsistent formatting of citations (e.g., author names in capital, erroneous use of brackets); some of these issues are highlighted below

1. P1, L17: "comparing" **[Fixed]**

2. P1, L22: "Yuste" **[Fixed]**

3. P3, L6: "nitrification rate. . . correlates" **[Fixed]**

4. P6, L1: if alpha refers to matric potential at maximum drainage, I am not sure I understand why $D_0$ (a function of alpha) refers to the modal rather than maximum pore throat diameter

   (a) This has been clarified as follows: "...is a parameter that indicates the matric potential at which the water retention curve exhibits the steepest slope".

The steepest slope of the curve implies that when a soil is subjected to progressively decreasing matric potential, the largest amount of water will be extracted at $\psi = -\alpha^{-1}$. This also implies that the corresponding pore size $D_0$ is the most common.

5. P6, L9: "top axis of the figure", which figure? I would refer to the figure number **[Fixed. Also the sentence was moved down so that it comes after Fig 1 was properly introduced]**

6. P6, L15: "unimodal" P6, L16: extra full stop? This sentence appears incomplete **[Fixed. The latter senetnce was fixed as: "SWC of soils that exhibit bimodal pore size distribution can be described by sums of two van Genuchten curves (Durner, 1994):"]**

7. P7, L12: check use of brackets - "Chowdhury et al. (2011b)" **[Fixed]**

8. P7, L16: "Watson" **[Fixed]**

9. P8, L4: this sentence appears incomplete **[Fixed as "But rather, its effect on SOM decomposition rate (dC/dt) is accounted for through its impact on the accessibility of SOC (Davidson et al., 2012). "]**

10. P11, L17: "important to note" **[Fixed]**

11. P14, L8: but in Figure 5, $k_{a,min} = 0.8$ as well P15, L4: what does "explained in its entirety" mean? Based on which performance metric? **[The statement in P14, L8 was corrected and now indicated the two values tested in the reported results. The sentence in P15, L4 refers to how the model works. It is not a general statement about moisture sensitivity. In the framework of the proposed model, there are no factors other those explained by the shape of SWC that can explain moisture sensitivity. The sentence was rephrased for calrity as "In the proposed model, sensitivity of SOM decomposition to soil**

moisture dynamics is explained in its entirety by the SWC, which directly dictates air content, water content and matric potential. "]

12. P15, L17: "soils that were. . ." **[The current phrasing is correct, see emphasized words here:"...individual *samples* of the same soil that *were* incubated at different levels...". No change was made.]**

13. P16: to avoid having incubation duration as a confounding factor, only the first data points from the Arnold et al. (2015) study could be used **[That would work if we were only looking for the optimal decomposition rate. But in this model, we also need to know the available SOC pool. Moreover, having multiple measurements over time increases the statistical robustness of the fitted parameter. No change was made.]**

14. P16, L21: more than inter-sample differences, the data from Miller et al. (2005) show strong Birch effect (Birch 1958) – longer dry periods trigger larger respiration pulses. This effect, which is widespread, cannot be captured by the proposed model. **[It is correct that the model does not account for wetting history. In the first wetting cycle, there should not be any difference of wetting-history between the 4-week and 2-week treatments. But, if you look closely at the data it clear that the 2-week rate is consistently lower than the 4-week rate. This can only be attributed to inter-sample differences. We provided two versions of models in which ignored or considered this difference. In both cases the Birch effect was not captured by the model. We added one statement to highlight this fact.]**

15. P17, L15: delete "in the" **[Fixed]**

16. P17-18: the structure of the Discussion and Conclusion section is a bit strange, with two introductory paragraphs and a single numbered subsection **[the headed subsection was unnecessary and is now removed.]**

17. P26, last line of the caption: "diameter" **[Fixed]**

18. Figure 2: check if labels (B) and (C) are correctly placed; the caption is not consistent with the figure and does not explain what panel (d) shows **[Fixed]**

19. P30, caption: no explanation of the difference between top and bottom panel is provided **[Explanation added.]**

20. Figure 6: check panel labels – now only (W), (I), and (D) appear as labels **[Explanation added.]**

21. Figure 7: not clear what is the difference between red and black curves **[The red curves were effective saturation curves (on secondary axes) that we plotted fo diagnostic purposes and were mean to be commented out in the code. They are now removed.]**

22. Figure A3: "bulk density" **[Fixed]**

23. Figure A4: what are the numbers in brackets? Is the number of significant digits reasonable? **[These are matric potential values predicted by pedotransfer function. The numbers are now reformatted.]**

**References provided by Reviewer #2**

1. Birch, H. F. 1958. The effect of soil drying on humus decomposition and nitrogen availability Plant and Soil 10:9-31.

2. Manzoni, S., F. Moyano, T. Kätterer, and J. Schimel. 2016. Modeling coupled enzymatic and solute transport controls on decomposition in drying soils. Soil Biology and Biochemistry 95:275-287.

3. Moyano, F. E., Vasi- Iyeva, N., and Menichetti, L.: Diffusion based modelling of temperature and mois- ture interactive effects on carbon fluxes of mineral soils, Biogeosciences Discuss., https://doi.org/10.5194/bg-2018-95, in review, 2018.

4. Tang, J. Y., and W. J. Riley. 2013. A total quasi-steady-state formulation of substrate uptake kinetics in complex networks and an example application to microbial litter decomposition. Biogeosciences 10:8329- 8351.

5. Yan, Z., Liu, C., Todd-Brown, K.E. et al. 2016. Pore-scale investigation on the response of heterotrophic respiration to moisture conditions in heterogeneous soils. Biogeochemistry 131: 121–134, https://doi.org/10.1007/s10533-016-0270-0

6. Yan, Z., B. Bond-Lamberty, K. E. Todd-Brown, V. L. Bailey, S. Li, C. Liu, and C. Liu. 2018. A moisture function of soil heterotrophic respiration that incorporates microscale pro- cesses. Nature communications 9:2562.

**References cited in response to Reviewer #2**

1. Durner, W. (1994), Hydraulic conductivity estimation for soils with heterogeneous pore structure, Water Resour. Res., 30, 211–223, doi:10.1029/93WR02676.

2. Iden, S. C., and W. Durner (2014), Comment on "Simple consistent models for water retention and hydraulic conductivity in the complete moisture range" by A. Peters, Water Resour. Res., 50, 7530–7534.

3. Peters, A. (2013), Simple consistent models for water retention and hydraulic conductivity in the complete moisture range, Water Resour. Res., 49, 6765–6780.

---

## Author Comment (AC3) · 17 Oct 2018

**Major Comments**

The manuscript proposes a new modeling framework that integrates the important role of soil water potential on regulating the rate of soil respiration. The model is built on assuming a single pool soil organic matter (SOM) where a first-order kinetics for the rate of SOM decomposition is considered. Authors have expanded the decay rate of SOM ($k$ parameter) to incorporate for the role of biophysical factors, mainly matric potential. This step is performed by a simple and testable exponential relationship between the decay rate and matric potential. The model is then expanded to include variations in oxygen and substrate diffusion as a function of matric potential and soil

depth. The simple nature of proposed mathematical framework allows its application for large-scale carbon cycle and climate models while preserving the effects of some of the key biophysical factors. This step is performed nicely in this model by reducing the number of calibration parameters and limiting them to some measurable quantities. The model is ultimately tested again good amount of datasets.

Overall, the technical quality of the manuscript is high and the proposed model has potential to be used in other biogeochemical gas flux models to account for the role of water content and potential, individually. I have some minor comments and recommendations that I believe could help the manuscript to be stronger and accessible for broader audiences.

**Comment 1:** My main suggestion is to better discuss uncertainties and limitations associated with the previously developed models that the current model aims to address those limitations. At the moment, it is not completely clear how incorporating matric potential into the model improves the model predictions compared to the models without this feature.

> Several changes that were in response to the other reviewers' comments will also address this issue. We have clarified what the scope and limitation of the proposed model are (see in particular Comment 7 of Reviewer #1 and Comments 1 and 2 of Reviewer #2). We added a new paragraph and figure added in the end to illustrate how moisture sensitivity curves vary by soil textural class (SWC parameters for the textural class averages were derived from ROSETTA pedotransfer function).

**Comment 2:** While the idea of using SWC is nice, the implementation and formulation is rather confusing and hard to follow. The main problem might be the inadequate description of the parameters and the links of parameters through the equations.

> There were some typographic errors in the main moisture sensitivity equation (Eq 12) that may have contributed to this lack of clarity. SWC contributes to moisture sensitivity in three ways: effect of water potential, effect of oxygen concentration, and effect of aqueous diffusion. The corrected Eq 12 now clearly shows this combined effect as a product of the three contributions (Eqs 7, 9, and 10, respectively) as explained by Eq. 6.

**Comment 3:** I also found that the manuscript is a bit bulky in the introduction and method descriptions. I suggest shortening the introduction and methods. While some of the discussions and examples in the introduction and method are informative, I think it might be destructing. For instance examples and discussions on nitrification process could be misleading, since the main story is about respiration and the connection between respiration and nitrification processes is not immediately clear even though both could be aerobic processes. If this part is necessary, I would suggest to provide a discussion on its need.

We agree that the introduction and methods are longer than typical. The current version of the manuscript evolved in response to feedbacks we received after presentations at AGU, EGU and other smaller venues. Because the main thesis of this research falls at the intersection soil biogeochemistry and soil physics, lack of adequate familiarity of concepts on both sides appeared to have been a roadblock in effectively communicating the main message. The discussion around nitrification was needed because Stark and Firestone (1995)–one of the key papers that we relied for developing the water-potential dependence–used activity of nitrifying bacteria as a model system. We added additional statement to clarify this: "They used nitrifying (ammonium oxidizing) bacteria as a model system, in which nitrification rate was considered as a surrogate for microbial activity."

**Comment 4:** My other suggestion is to better explain the difference between water content and matric potential, maybe in a schematic. For instance the independent

relationship of water potential from water content and its effects on osmotic potential that is discussed in the manuscript is not so clear. This is important motivation of the paper and could be illustrated a little bit more. Meanwhile, the effects of osmotic potential are discussed in the introduction, but its incorporation in the model is not so clear, even though it has been assumed that Eq. 6 could also account for osmotic potential.

We added clarifying sentences and phrases in the introduction and the methods sections to this effect.

**Minor comments:**

1. Page 1, Line 21: "are strongly correlated heterotrophic respiration rates" grammar error? References are not consistent. Some author names are capital and some are not. **[We added the missing preposition 'with'. The citation database was updated so that all names are capitalized consistently.]**

2. Page 2, line 3: "films is dependent" grammar? "Moisture sensitivity curve" is probably not accurate terminology. I suggest to define moisture sensitivity term. **[The incorrect verb was fixed. The term 'moisture sensitivity' curve has been used by others as well (e.g., Lawrence, C. R., Neff, J. C. and Schimel, J. P.: Does adding microbial mechanisms of decomposition improve soil organic matter models? A comparison of four models using data from a pulsed rewetting experiment, Soil Biol Biochem Soil Biol Biochem, 41(9), 1923–1934, 2009).**

3. In page 4 line 18, "biophysical rates". Here it is not clear what authors mean. **[Corrected as "biophysical factors"].**

4. In Eq. 6 and 7 different k parameters are used. I would suggest to better define

these parameters. The current version is a bit confusing. **[More explanations given as suggested]**.

5. Section 2.2 "SOM dynamics modeling" is very long that makes it hard to read and follow the method. I suggest breaking down this section into subsections with detailed subheadings. **[Subheadings were added as suggested]**.

6. In line 6, page 8, I think the difference between gas and liquid diffusion coefficients of oxygen is about 4 orders of magnitude, I suggest checking the number, once more. **[Corrected as suggested]**.

7. Figure A1 is unclear. At the moment, it is unclear what dashed lines mean. PWP and FC could be defined in the caption of the figure. "Bioavailable SOC" should be defined. The term has not been defined and discussed in the rest of the manuscript. **[We added "The dashed-lines of the Franzluebbers soils denote compressed samples." Also we defined PWP and FC]**

---

## Author Response (AR2)

**Respones to Reviewer Comments: "On the role of soil water retention characteristic on aerobic microbial respiration"**

by T.A. Ghezzehei, B. Sulman, C.L. Arnold, N.A. Bogie, A.A. Berhe

December 29, 2018

We are grateful for all the constructive comments and suggestions we received. Reviewer #1 recommended to accept the manuscript in its present form. Therefore, no changes were made to address their feedback. Detailed responses to reviewers #2 and #3 are provided below. Responses to major comments are shown as block quotes (indented on both sides). Responses to minor comments are added in **bold** font next to the comments. Changes we made in response to specific comments are quoted below each response as well.

We believe the manuscript is now much clearer and stronger because of these reviews. Thank you!

**REVIEWER #2**

**Major Comments**

This paper presents a method for modeling the effects of soil hydrology on soil decomposition rates. The authors do a good job of integrating different water related processes that may limit decomposition into one relatively simple equation. Their approach is therefore more mechanistic than conventional approaches. While the different processes they consider have been explored in previous studies, this study's approach and results are unique in the way the soil water retention characteristics are used to simulate a more complete combined effect of water potential and water content.

I think the manuscript only needs some minor corrections before publication. But I also have some suggestions and comments that the authors may want to consider to improve the final paper.

**Comment 1:**In Eq. (9), you make oxygen limitation proportional the the relative diffusivity of O2. While this seems like a practical solution, it ignores the fact that a limitation is not only a function of the supply but also of the demand of O2. This means your model may simulate an O2 limitation when none actually exists. Although this may be beyond the scope of your model, it may be good to mention it in the discussion as a potential limitation.

> **Response 1**: Thank you for this important observation. You underscored an important aspect of the variable $\kappa_{a.\min}$, which we understood intuitively but have struggled to explain. The revised paragraph that follows Eq. (9) is much more nuanced.
> "The parameter $\kappa_{a.\min}$ represents the minimum relative SOM decomposition rate when the soil is fully saturated and the O2 limitation supply rate is at its lowest. A value of $\kappa_{a.\min} \approx 1$ implies no that no $O_2$ limitation would occur even when the local supply rate is at its lowest. One possible cause for such phenomenon could be inherently low $O_2$ demand because of other limiting factors (e.g., lack of essential nutrients or presence of inhibiting factors). It is also reasonable to expect high values of $\kappa_{a.\min}$ for well aerated conditions (e.g., shallow soil depth or small samples), in which gaseous $O_2$ replenishment occurs readily. In contrast, when the inherent respiration rate is high (e.g., substrate and essential nutrients are abundant and minimal inhibiting factors exist) or rate of replenishment is slow (e.g., deep within soil profile) the value of $\kappa_{a.\min}$ is expected to approach zero. Further controlled experiments are needed to better constrain how this parameter varies with depth or the inherent $O_2$ demand of soils. The effect of on the overall trend of the relative decomposition rate is illustrated in Fig 2a. "

**Comment 2:**You cite Davidson 2012 who made diffusivity a cubic function of soil water content, but here you make it a function of tortuosity (Eq. 10), i.e. to the power of 1/2. Both have theoretical or empirical evidence to back up the approach, but some justification in the paper of why the tortuosity is preferred over a stronger exponential function often obtained for diffusion in porous media would be helpful.

> **Response 2**: We cited Davidson et al (2012) because they too have considered substrate accessibility as a distinct process that

limits respiration. Over the past several decades numerous formulas of tortuosity have been proposed to scale aqueous and gaseous diffusivities; the models of Millington-Quirk and Moldrup et al being some of the most notable in the soil physics literature. No single model is superior over the others across many soil types. In the end, diffusivity model that works best for the particular soil of interest would more appropriate. Because the main goal of this study was to investigate the role of SWC, we decided to keep all the other parameters consistent across all the soils we tested. It appeared to us that a mild power of 1/2 (Bruggeman model) would serve better for the wide range of soils we considered here. It is important to note that the framework that we proposed (Eq. 12) can be updated with soil specific parameters; including the the powers of the aqueous and gaseous phase tortuosity factors as well as of the parameter $\lambda$, which describes the sensitivity of respiration to matric potential.

**Comment 3:** In the discussion it would be very informative to compare the model results here (as shown in figure 11) with the data-driven results from Moyano et al. 2012.

**Thank you for this recommendation. We added the following paragraph at the end of the discussion.**
It is informative to compare the above results with data-driven (statistical) moisture sensitivity functions derived by Moyano et al, (2012) (c.f. their Fig 3). In both models, the matric potential at peak relative respiration decreases (becomes more negative) with increasing clay content (degree of textural fineness). However, the models differ in their prediction of the range of matric potential at which respiration remains elevated. Our model shows distinct peak of respiration over very narrow range for the two coarsest textures (Sand and Loam Sand), whereas the curves predicted by Moyano et al (2012) have similar overall pattern across the entire range of clay content analyzed. The most remarkable difference appears in the moisture sensitivity functions expressed as functions of relative saturation. The model of Moyano et al. (2012) predicts that respiration decreases nearly linearly until the soils are completely dry (saturation =0). Whereas our model suggests that respiration ceases when the saturation drops to residual moisture content. The prediction is based on

the fact that in very dry soils, the matric potential decreases very rapidly with very small decrease in water content. Therefore, the prediction of our model in the dry end appears sound. Another distinction between the two models is related to the role of SOM in moisture sensitivity. Although our model does not directly incorporate the effect of SOM in moisture sensitivity, the effect of SOM or SWC would also be reflected as variation in moisture sensitivity with change SOM. In contrast, the model of Moyano et al. (2012) predicts no effect of SOM moisture sensitivity function.

**Comment 4:** Throughout the document, articles and prepositions are missing in several places.

We carefully read the manuscript and added the missing articles and prepositions.

**Minor comments**

1. P2L8: the meaning of "average" here is not clear. Averaged over what?
   **We revised it as:**
   "...corresponding volumetric-average matric potential"

2. P3L20-21: Matric potential was defined earlier so shouldn't need further explanation. In addition, this sentence is inaccurate (WP is the result of..., not comprised of...).
   **The redundant sentence was deleted.**

3. P4L1: the term "reducing" may be differently interpreted, either as becoming more negative or approaching 0 (if thinking in absolute quantities). I think that using "more negative", here and elsewhere, is clearer.
   **We added the following explanation in parentheses:**
   "(changing towards larger negative values)"

4. P5L1-2: The issue of variability in soil properties should also be mentioned as a main factor.
   **Yes, soil heterogeneity is one factor as well. But it is not one of the factors that this proposed model aims to address. No change was made in response to this question.**

5. P5L16: SWC does not "exert direct control" on anything. It is simply the relationship between the stated factors.
**We rephrased it as:**
"This soil-specific relationship determines macroscopic and microscopic water content distributions, and indirectly influences flow of water, transport of dissolved constituents and gas fluxes."

6. P5L18-19: There is repetition inside this paragraph and with the intro section.
**A redundant sentence was removed and the other sentences were shortened to avoid repetition.**

7. P6L6: "can be re-written"
**Corrected.**

8. P6L10: Pore throat diameter is defined both as $4\sigma/\psi$ and $4\alpha\sigma$, although one is defined as the model pore throat diameter. This is confusing and seems inconsistent.
**The former is definition of pore-throat diameter using Young-Laplace equation. The latter is the mode of the pore size distribution, which corresponds to matric potential of $\psi = 1/\alpha$. We eliminated the potentially confusing term 'modal' and rewrote the sentence as:**
"... and $D_0 \approx 4\alpha\sigma$ is the mode of the pore-throat diameter distribution"

9. P8L7: a steeper
**Corrected as part of broader revision of the paragraph. See response to comment #5 of Reviewer #3.**

10. P9L23: you mean Eq.(7) ?
**Yes. Corrected**

11. P11L2: "mechanism through which water"
**The sentence was revised. See response to comment #6 of reviewer #3.**

12. P11L22: Why do you use tau (which is nowhere defined) and not t at the end of Eq.(11)?
**The variable $\tau$ is dummy variable of integration that stands for $t$. It is eliminated when the integration is carried out. It is used because the symbol $t$ cannot be both variable of**

integration and limit of the integral. It is now explained after the equation. See here for detailed explanation: http://mathworld.wolfram.com/DummyVariable.html.

13. P12L1: for consistency with the new version of Eq(11), update the expression in this line, i.e. make as functions of time.
**Added.**

14. P16L3: Stay consistent with naming the soils, either using the study names (Arnold, etc) or meadows, etc, reflecting headers in Methods section.
**We added the study name in parentheses when the soils are described as meadow or mineral soils. References such as "SOM-rich meadow" are helpful in this case to help explain differences of the moisture sensitivity in relation to the "mineral soils".**

15. P16L4: In Fig. 6, add in caption what lines represent. Also, caption mentioned Yosemite, which is nowhere else.
**Replaced with 'from SOM-rich meadow soils (Arnold et al., 2015)' to be consistent wit the rest of the manuscript.**

16. P16L22-P17L3: I think the interpretation here is wrong. Soils with higher SOC are likely to result in a higher $C_0$ (initial C). At the same time a higher $C_0$ correlates with a lower overall C quality (lower $K_0$). The incubation duration can affect these numbers. But the duration was equal for all samples of Arnold and therefore does not explain the correlation in panel b.
**Thank you for this important observation. In light of the revision part (c) of Fig 8 was deemed unnecessary and removed. The paragraph was revised as follows.**
"The best-fit optimal decay rates for all the steady-moisture experiments are plotted against (a) SOC and (b) the initial active-fraction of SOC ($C_0$) in Fig 8. The overall SOC decomposition rate ($\log(\kappa_\circ)$) was negatively correlated with both $\log(SOC)$ and $\log C_0$. These correlations suggest that, in the long run, accumulation of high SOC leaves behind C that recycles at increasingly slower rate. Furthermore, comparing Figs 8a and 8b suggests that soils with higher SOC are also likely to have higher proportion in the active pool ($C_0$). However, these interpretations should be taken with caution considering the duration of the incubation experiments of Franzluebbers (Franzluebbers,

1999) and Don (Moyano et al., 2012) were much shorter than that of Arnold et al. (2015) (24 and 31 days vs 395 days, respectively)."

17. P17L10: check caption in Fig.10 for consistency.
    **The caption was revised as:**
    "Model evaluation by comparison with experiment of Miller et al. (2005) under drying and rapid-wetting conditions (2-week and 4-week intervals): (a) observed dynamics of water holding capacity; (b) comparison of measured $CO_2$ efflux with model prediction assuming identical optimal decomposition rate $\kappa_\circ$ for both wetting intervals; and (c) same as above but with different $\kappa_\circ$ parameters for the two wetting intervals."

18. P17L20: I don't see why this sentence is necessary. Having the SWC parameters, water content could be just as well taken as the independent variable.
    **Yes, given SWC $\theta$ and $\psi$ are interchangeable. But we found such a reminder at the outset to be helpful when presenting this work to audiences accustomed to seeing moisture sensitivity with $\theta$ or $\theta/\phi$ as the independent variable (shown on the abscissa). Also, we felt that $\psi$ is a more appropriate variable for comparing different soils is not always obvious.**

19. P17L24: Replace "However" with "Thus"
    **Corrected as suggested.**

20. P18L9 replace "under" with "when approaching"
    **Corrected as suggested.**

21. P19L5-10: Sensitivities falling to 0 at water contents above 0.2 are not a common observation.
    **We agree that such observations are not common. However, most observations are based on lab cores or near surface soil sensors. However, it is plausible and the model indirectly suggests that at sufficiently large depth within soil profile the sensitivity can drop to zero at effective saturation levels of 80% or more. Note that it does not reach zero in the examples shown in Fig 11.**

22. P19L17-18: Describing this study as a first step is inaccurate. This is another step within many studies in this topic.

**Correct. It was written to imply 'the first step for us in this line of inquiry'. We replaced 'first step' by 'one approach'**

23. P20L23: remove the subjective qualifier "particularly valuable"
    **Revised as suggested.**

**Updated Reference provided by Reviewer #2**

1. Moyano, F. E., Vasilyeva, N., and Menichetti, L. (2018). Diffusion limitations and Michaelis-Menten kinetics as drivers of combined temperature and moisture effects on carbon fluxes of mineral soils. Biogeosciences, 15(16), 5031-5045. https://doi.org/10.5194/bg-15-5031-2018

**REVIEWER #3**

This manuscript develops a simple but physically-based model of soil water limitation on SOM decomposition. It fits the initial C content and decay rate to the model and tests against four datasets over a wide range of soil textures and water potential. I think this model represents an improvement over other soil water decomposition relationships that exist since it only has one tunable parameter ka,min. It also implicitly incorporates the effect of aggregation on pore size distribution. Earth system modelers could fairly easily substitute this model for older scalar modifiers of the decomposition rate based on soil water potential.

**Comments**

1. P3, L15: I've always taken kind of a semantic issue ascribing high intracellular osmotic potential to "accumulating electrolytes and solutes". Water could also just be exiting the cell due to the existing gradient.
   **Correct. However, release of water to maintain osmotic equilibrium with the outside can only be sustained if the loss of the accompanying loss turgidity is not significant. The statement was meant to highlight that managing osmotic stress requires metabolic cost (accumulation of osmolites) or loss of water as the reviewer suggests (presumably with loss of activity as well). To show that this is not the only response, we revised the sentence to read as:**
   "...microorganisms can react to increasing osmotic stress (...) by accumulating electrolytes... "

2. P4, L2: Consistency of terms: At the bottom of the previous page, water content is described parenthetically as matric potential. I like framing the various effects of water by the terms you use to describe the parts of the model: matric potential, dissolved oxygen, and substrate accessibility. This seems a little clearer to me than water potential, soil aeration, and water content (P4, L16), but it depend on the field.
   **We agree. We revised as suggested.**

3. P7, L4: After reading the reviewer comments about the possible bimodality of the other datasets (something I also noticed), I understand why this paragraph is here, but I dont think the readers will make the connection unless you use this paragraph as a direct justification for

assigning two of the datasets to be unimodal.

**We added the following clarification after introduction of the bimodal equation:**

"The bimodal curve was used only for soils that exhibited rapid drop in water content with the application of low suction, which is characteristic feature of structured soils. ". In section 2.3 after introducing the Arnold et al dataset we added: "All the SWC data were fitted with bimodal SWC model of (Durner, 1994) because they exhibited the characteristic rapid decrease in water content at low suction."

4. P8, L7: They observed that steeper should be They observed a steeper?
   **Corrected as part of revision in response to comment #5.**

5. P8, first paragraph: I think this paragraph is trying to convince me that water potential and diffusion limitation are both important. But I feel like it goes back and forth between water potential is important without diffusion limitation and both are important, maybe you can cut text here a little for a simpler point.
   **The paragraph was revised as follows:**
   "Previously, Stark and Firestone (1995) used two independent techniques to evaluate (a) the relative importance of water potential on cytoplasmic dehydration and (b) the role of water content diffusional limitations in controlling soil microbial activity. They used nitrifying (ammonium oxidizing) bacteria as a model system, in which nitrification rate was considered as a surrogate for microbial activity. In the first experiment, they used well mixed soil slurries, in which NH4 was maintained at high concentrations and osmotic potential was regulated by the addition of K2SO4. In a companion experiment, they incubated moist soils at wide range of matric potential and kept N level elevated by the addition of NH3 gas. In the former experiment, nitrification rate declined exponentially with reduction in water potential of the slurries (0 to  -4000 kPa). The latter experiment exhibited steeper decline of nitrification across the same range of total water potential as the first experiment, demonstrating that diffusional limitation exacerbates cytoplasmic-dehydration effect of lower total water potential. Similarly, (Tresner and Hayes, 1971) showed that in the absence of diffusion limitation the survival probability of fungi declines with water potential. Therefore, we treat the effects of (a) reduced diffusion (which depends on water content) on accessibility of SOC (Davidson et al., 2012). and (b) reduced matric potential on

cytoplasmic-dehydration as distinct interacting factors."

6. P11, L1: Check the grammar in this first sentence.
   **We revised it as:**
   "Water content also determines substrate accessibility to decomposer microorganisms, thereby influencing rate of SOM decomposition."

7. P19, L20: What study was this? Peat was not mentioned in the study descriptions.
   **We added the following to the description of soils from Arnold et al:**
   "The wet meadow soils were classified as a fibrous peat in the surface layer, but the intermediate and dry locations had mineral soils with high organic matter content.". We also revised the conclusion to "... peat and organic soils (Arnold et al., 2015),...".

8. Table 1: Don study says it lasted 30 days in text, but 1 is listed in the table. Also, why 100? Were these data already aggregated or something?
   **The correct numbers are 32 days and 767 measurements (hourly). The question mark next to this dataset was meant to serve as a reminder to update the placeholder numbers, but we did not catch it. Thank you!**

9. In Figure 4, the unimodal model is defined as Van Genuchten, but in Figure 1 the Van Genuchten model is bimodal. Typo?
   **The incorrect legend of Fig 1 was fixed.**

Manuscript prepared for submission to EGU Biogeosciences          **Deleted:** 29-Dec-1823-Dec-1821-Dec-18

[revised manuscript text omitted]